# Genetic inhibition of CARD9 accelerates the development of atherosclerosis in mice through CD36 dependent-defective autophagy

Yujiao Zhang [1], Marie Vandestienne [1,19], Jean-Rémi Lavillegrand [1,19], Jeremie Joffre[1,2], Icia Santos-Zas[1], Aonghus Lavelle[2,3], Xiaodan Zhong[1], Wilfried Le Goff [4], Maryse Guérin[4], Rida Al-Rifai [1], Ludivine Laurans[1], Patrick Bruneval[1,5], Coralie Guérin [6], Marc Diedisheim [7], Melanie Migaud [8], Anne Puel[8,9], Fanny Lanternier[8], Jean-Laurent Casanova[8], Clément Cochain[10,11], Alma Zernecke[11], Antoine-Emmanuel Saliba [12], Michal Mokry [13], Jean-Sebastien Silvestre [1], Alain Tedgui [1], Ziad Mallat[1,14], Soraya Taleb [1], Olivia Lenoir [1], Cécile Vindis[15], Stéphane M. Camus [1], Harry Sokol [2,3,16,17] & Hafid Ait-Oufella [1,2,18] ✉

Caspase recruitment-domain containing protein 9 (CARD9) is a key signaling pathway in macrophages but its role in atherosclerosis is still poorly understood. Global deletion of *Card9* in *Apoe*[-/-] mice as well as hematopoietic deletion in *Ldlr*[-/-] mice increases atherosclerosis. The acceleration of atherosclerosis is also observed in *Apoe*[-/-]*Rag2*[-/-]*Card9*[-/-] mice, ruling out a role for the adaptive immune system in the vascular phenotype of *Card9* deficient mice. *Card9* deficiency alters macrophage phenotype through CD36 over-expression with increased IL-1β production, increased lipid uptake, higher cell death susceptibility and defective autophagy. Rapamycin or metformin, two autophagy inducers, abolish intracellular lipid overload, restore macrophage survival and autophagy flux in vitro and finally abolish the pro-atherogenic effects of *Card9* deficiency in vivo. Transcriptomic analysis of human *CARD9*-deficient monocytes confirms the pathogenic signature identified in murine models. In summary, CARD9 is a key protective pathway in atherosclerosis, modulating macrophage CD36-dependent inflammatory responses, lipid uptake and autophagy.

Atherosclerosis is a chronic inflammatory disease of large and medium-sized arteries that develops in response to subendothelial retention and modification of ApoB containing low-density lipoproteins (LDL)[1]. Inflammatory activation of endothelial cells orchestrates the recruitment of different subsets of circulating leukocytes, notably monocytes, into the vascular wall. Recruited monocytes significantly contribute to the pool of intimal macrophages[2], which promote the growth of atherosclerotic plaque after differentiation, activation and proliferation. Macrophage activation, an instrumental step in the development of atherosclerosis, is associated with the upregulation of pattern recognition receptors for innate immunity, including scavenger receptors (SR-A, CD36) and pathogen recognition receptors,

such as Toll-like receptors (TLRs) and Dectin receptors[3–5]. A broad range of molecules and particles bearing danger-associated molecular patterns, including oxidized LDL (oxLDL), can be taken up by macrophages, leading ultimately to the formation of pro-inflammatory foam cells[6]. CD36 is an archetypal pattern recognition receptor that binds polyanionic ligands of both pathogen- and self-origin[7] including oxLDL[8]. CD36 forms a complex with the TLR TLR4-TLR6 heterodimer, which recognizes oxLDL and stimulates pro-inflammatory pathways, including the NLRP3 inflammasome activation responsible for proIL-1β. cleavage and IL-1β secretion[9]. A large body of evidence suggests that cumulative metabolic/inflammatory signals and impaired efferocytosis foster apoptosis and secondary necrosis of foam cells, which contributes to the growth of the necrotic core and progression of atherosclerosis[10]. However, the critical downstream pathways that drive both macrophage activation and conversion into foam cells are still poorly understood.

Here, we investigated the role of Caspase recruitment-domain containing protein 9 (Card9), a key regulator of inflammation, in atherosclerosis. Card9 is an adapter protein that integrates pattern recognition receptor downstream signals in macrophages and dendritic cells[11]. Card9 is particularly involved in response to fungi via C-type lectin sensing, but also in response to bacteria by mediating nucleotide-binding oligomerization domain 2 (NOD2)-dependent p38/JNK signaling and TLR signaling[12]. Card9 is required to mount appropriate immune responses through the production of interleukin (IL)−6, IL-17A, IFN-γ, and IL-22, which can affect gut microbiota[13], with potential impact on atherosclerosis[14]. The role of Card9 in inflammatory diseases is ambiguous, being pathogenic in post-ischemic cardiac remodeling[15], but protective in experimental colitis[13]. In two recent studies, the role of CARD9 in atherosclerosis has been explored, but results reported contradictory findings. One study showed increased lesion size in chimeric *Ldlr*/*Card9*−/− mice[16], whereas the other found that deletion of haematopoietic Card9 did not affect atherosclerosis in chimeric *Ldlr*−/− mice under hyperglycaemic conditions[17]. Moreover, the underlying mechanisms linking Card9 engagement to atherosclerosis development remain largely unknown.

Here, using several complementary approaches and state-of-the-art models, we show that Card9 signaling pathway in macrophages regulates cytokine production, lipid upload, and cell survival. Global, as well as hematopoietic deletion of Card9, markedly accelerates the development of atherosclerosis, independently of the adaptive immune system. Mechanisms of the pro-atherogenic effects of Card9 deficiency mainly involve CD36-dependent defective autophagy.

## Results

### Genetic invalidation of *Card9* accelerates atherosclerosis in *Apoe*−/− mice

To gain insight into the immune cells expressing *Card9*, single-cell analysis of total cells from mouse atherosclerotic aortas (see methods) was performed (Fig. 1A, B and Supplementary Fig. 1). *Card9* transcript was detected in myeloid cells, and macrophages in particular, including atherosclerosis-associated inflammatory and *Trem2*hi/Foamy macrophages (Fig. 1A, B and Supplementary Fig. 1a, b). Analysis of a second, independent scRNA-seq dataset of *Ldlr*−/− mouse atherosclerotic aortas[18] corroborated preferential detection of *Card9* transcripts in myeloid cells, including macrophages (Supplementary Fig. 2). Immunofluorescence staining confirmed that Card9 was expressed in atherosclerotic lesions of *Apoe*−/− mice at both early (Supplementary Fig. 3a) and advanced stages of atherosclerosis (Fig. 1C), and mainly co-localized with MOMA+ macrophages (Fig. 1C). In vitro, Card9 expression in macrophages was induced by oxLDL (Supplementary Fig. 3b).

To investigate the role of Card9 in this experimental setting, we generated *Apoe*−/−*Card9*−/− mice. *Card9* deficiency was confirmed by qPCR in peritoneal macrophages (Fig. 1D) and by immunostaining of atherosclerotic plaques (Supplementary Fig. 4a). Body weight was slightly increased in *Card9*-deficient mice, but no significant differences in plasma cholesterol levels were observed between male *Apoe*−/− *Card9*+/+ and *Apoe*−/−*Card9*−/− mice (Supplementary Fig. 4b, c). At 8 weeks of age, animals were put on a high-fat diet for 6 weeks to accelerate plaque formation. *Apoe*−/−*Card9*−/− mice showed a significant increase in atherosclerotic lesion size in the aortic sinus ($368 \pm 64$ vs $278 \pm 87$. $10^3$ μm², $P < 0.05$; Fig. 1E).

*Card9* deletion in *Apoe*−/− mice induced a switch toward a more inflammatory plaque phenotype with a significant increase in macrophage accumulation (Fig. 1F) and necrotic core size (Fig. 1G). Collagen content was increased in plaques of *Apoe*−/−*Card9*−/− mice (Supplementary Fig. 4d), but T cell accumulation was similar (Supplementary Fig. 4e).

### Dampened systemic pro-inflammatory cytokine signature in *Apoe*−/−*Card9*−/− mice

Because Card9 is known to modulate cytokine production and T cell polarization[13], we next investigated the immuno-inflammatory response in male *Apoe*−/−*Card9*+/+ and *Apoe*−/−*Card9*−/− mice. Leukocyte populations were analyzed by flow cytometry in both blood and spleen at sacrifice. We did not observe any significant difference in leukocyte percentages in blood between groups. We only found a slight increase in neutrophil and classical monocyte counts in the blood of *Apoe*−/−*Card9*−/− mice, compared to control *Apoe*−/−*Card9*+/+ mice (Supplementary Fig. 5). Splenocyte number was significantly higher in *Card9*-deficient mice but the proportion of myeloid and lymphoid populations was not different between groups (Supplementary Fig. 6). Splenocytes from *Apoe*−/−*Card9*−/− mice stimulated with IFN-γ and LPS produced less TNF-α (Fig. 1H) than those from control mice, but the production of IL-10 and IL-1β was not different. We then purified splenic CD4+ T cells from control *Apoe*−/−*Card9*+/+ and *Apoe*−/−*Card9*−/− mice and performed functional tests. In vitro, the proliferation of CD4+ T cells from *Apoe*−/−*Card9*−/− mice was significantly increased compared with control cells (Data non shown), and their production of IFN-γ and IL-17A was increased. There were no differences in IL-10 and IL-22 production (Supplementary Fig. 4f).

### Hematopoietic *Card9* deficiency increases atherosclerosis in *Ldlr*−/− mice

To confirm our result in another mouse model of atherosclerosis and to specifically investigate the role of hematopoietic Card9, we performed bone marrow (BM) transplantation experiments using either *Card9*+/+ or *Card9*−/− BM cells to repopulate lethally irradiated female *Ldlr*−/− mice. We confirmed that *Card9* gene expression was almost abolished in peritoneal macrophages of chimeric *Ldlr*−/−*Card9*−/− (Fig. 2A). Chimeric *Ldlr*−/− mice were then fed a high-fat diet for 8 weeks. We observed no difference in body weights (Fig. 2B) or serum cholesterol levels (Fig. 2C) between the 2 groups of chimeric mice. In blood, myeloid populations were not different between groups, but we observed a significant increase in circulating CD4+ T and B cell counts in chimeric *Ldlr*−/−*Card9*−/− mice (Supplementary Fig. 7). As shown in Fig. 2, hematopoietic *Card9* deficiency was associated with a significant increase in lesion development compared with controls, in the thoracoabdominal aorta (Fig. 2D), and in the aortic sinus (Fig. 2E). In addition, *Card9* deletion induced a more inflammatory plaque phenotype with a significant increase in both macrophage accumulation (Fig. 2F) and necrotic core size (Fig. 2G). We also observed an increase in collagen content in plaques of chimeric *Ldlr*−/−*Card9*−/− mice (Supplementary Fig. 8a). T cell accumulation in plaques was similar in the 2 groups (Supplementary Fig. 8b). Stimulated splenocytes from *Ldlr*−/−*Card9*−/− mice produced less TNF-α, IL-1β, and IL-10 than splenocytes from *Ldlr*−/−*Card9*+/+ mice (Fig. 2H). Splenic CD4+ T cells isolated from *Ldlr*−/−*Card9*−/− mice produced less IL-17A than CD4+ T cells from control *Ldlr*−/−*Card9*+/+ mice. IL-10, IFN-γ and IL-22 production was not affected in our experimental conditions (Supplementary Fig. 8c).

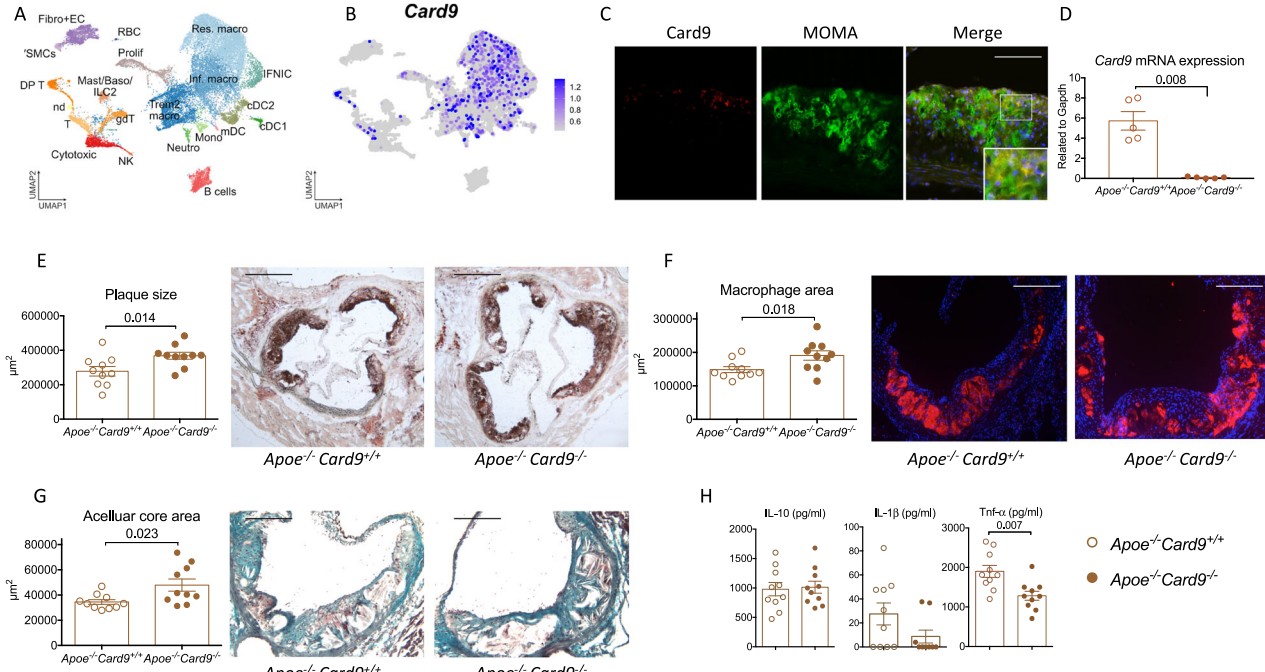

**Fig. 1 | Global Card9 deficiency accelerates atherosclerosis in Apoe-/- mice.**
scRNA-seq analysis of *Card9* transcript expression in murine arteries with 26,910 immune and non immune cells from the aorta of atherosclerotic mice (integrated analysis of 13 datasets). **A** UMAP representation of single-cell RNA-seq gene expression data and cellular lineage identification. (VSMCs vascular smooth muscle cells, Fibro+EC fibroblasts and endothelial cells, DP T double poisitive T cells, RBC red blood cells, Prolif proliferating cells, Mast/BAso/ILC2 mast cells, basophils and type 2 innate lymphoid cells, T T cells, NK natural killer cells, gdT gammadelta T cells, Neutro neutrophils, Mono monocytes, mDC mature dendritic cells, cDC1 and 2 classical dendritic cells 1 and 2, IFNIC type I interferon inducible cells, Trem2 macro *Trem2hi*/Foamy macrophages, Inf. Macro inflammatory macrophages, Res. Macro resident macrophages, nd not determined). **B** Card9 expression in single cells projected onto the UMAP plot. **C** Card9 (Red) and MOMA-2 (Green) immunofluorescent staining in plaques of 20-week old *Apoe-/-* mice (Representative of 7 atherosclerotic plaques). Scale bar 50 μm. **D** *Card9* mRNA expression by peritoneal

macrophages isolated from *Apoe-/-Card9+/+* and *Apoe-/-Card9-/-* mice (n = 5/group). **E** representative photomicrographs and quantitative analysis of atherosclerotic lesions in the aortic sinus of male *Apoe-/-Card9+/+* and *Apoe-/-Card9-/-* mice after 6 weeks of fat diet (2 experiments pooled, n = 10/group); Scale bar 200 μm. **F** representative photomicrographs and quantitative analysis of macrophage accumulation (MOMA staining, red) in atherosclerotic lesions of *Apoe-/-Card9+/+* and *Apoe-/-Card9-/-* mice after 6 weeks of fat diet (2 experiments pooled, n = 10/group); Scale bar 100 μm. **G** Representative photomicrographs and quantitative analysis of acellular area (Masson's Trichrome) of *Apoe-/-Card9+/+* and *Apoe-/-Card9-/-* mice after 6 weeks of fat diet (2 experiments pooled, n = 10/group); Scale bar 100 μm. **H** Cytokine production (ELISA in the supernatant) by Lps/Ifnγ-stimulated splenocytes from *Apoe-/-Card9+/+* and *Apoe-/-Card9-/-* mice (n = 10/group). Data are presented as mean values ±SD. Two-tailed Mann–Whitney test. Source data are provided as a Source Data file.

## Gut microbiota unlikely contributed to the acceleration of atherosclerosis induced by Card9 deficiency

Previous studies have shown that Card9 plays a critical role in gut microbiota homeostasis. *Card9-/-* mice have been shown to display dysbiosis[19], which has been implicated in atherosclerosis development[20]. To evaluate a potential impact of Card9-induced dysbiosis in our experimental conditions, we analyzed the bacterial microbiota was performed using 16S rRNA based sequencing. While there were no significant differences in alpha diversity between *Apoe-/-Card9+/+* and *Apoe-/-Card9-/-* mice (Supplementary Fig. 9a, b), beta diversity analysis showed a significant difference between the 2 groups, as demonstrated by the PCoA plot of Bray-Curtis distance (Supplementary Fig. 9c). To determine which taxonomic groups accounted for these differences, we performed linear discriminant analysis with effect size (Lefse)[21]. Compared to *Apoe-/-Card9+/+* mice, *Apoe-/-Card9-/-* mice displayed an increase in the pathobiont *Helicobacter* with a concomitant decrease in beneficial members of the *Firmicutes* phylum, including the order *Clostridiales*, as well as in *Candidatus arthromitus*, segmented filamentous bacteria. These bacteria are essential for Th17 maturation in the murine gut[22], and in the genus *Akkermansia*, a genus associated with a lean body type and favorable metabolic outcomes[23] (Supplementary Fig. 9d).

Given the marked dysbiosis in *Card9*-deficient *Apoe-/-* mice, we aimed to determine whether this dysbiosis was also observed in *Ldlr-/-* mice transplanted with *Card9-/-* bone marrow cells. As found in *Apoe-/-*

mice, 16S rRNA based sequencing showed no significant difference in alpha diversity among the 2 groups, but the beta diversity analysis showed significant differences in mice transplanted with *Card9-/-* BM cells (Supplementary Fig. 9e, f). Of note, because of transient aplasia and increased risk of sepsis chimeric *Ldlr-/-* mice received antibiotics during 14 days following lethal irradiation and BM cell transplantation. The administration of antibiotics caused marked changes in the microbiota in the 2 groups. In particular, at genus level, *Parasutterella* was increased in *Ldlr-/-Card9-/-* mice and the sulfate-reducing bacteria *Desulfovibrio* was enhanced in *Ldlr-/-Card9+/+* mice. Only one bacterial family, *Clostridiaceae_1* that was increased in *Ldlr-/-Card9+/+* mice, was concordantly altered in the two sets of experiments (Supplementary Fig. 9g–h). Taken together, these data confirm an effect of the Card9 on the gut microbiota composition. However, the highly divergent microbiota composition between *Apoe-/-* and *Ldlr/-* genetic backgrounds despite similar *Card9* effects on atherosclerosis suggests that the gut microbiota was unlikely involved in the vascular phenotype induced by *Card9* deficiency.

## Pro-atherogenic effect of Card9 deficiency is not dependent on adaptive immunity

As described above, *Card9* deficiency had major effects on CD4+ T cell proliferation and polarization. In order to evaluate the role of adaptive immunity in the acceleration of atherosclerosis observed in *Card9*-deficient mice, we backcrossed *Apoe-/-Rag2-/-* mice with *Rag2-/-Card9-/-* to

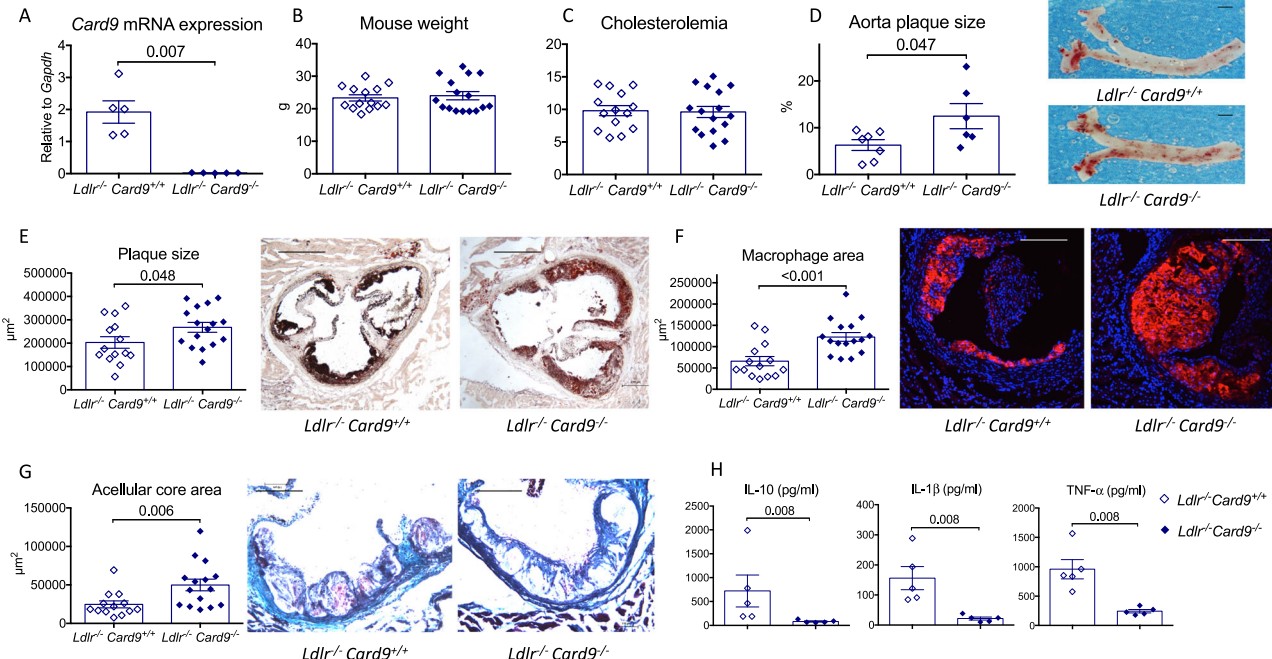

**Fig. 2 | Hematopoietic *Card9* deficiency accelerates atherosclerosis in *Ldlr*[-/-] mice. A** *Card9* mRNA expression by peritoneal macrophages isolated from chimeric *Ldlr*[-/-]*Card9*[+/+] and *Ldlr*[-/-]*Card9*[-/-] mice (*n* = 5/group). **B** body weight at sacrifice after 8 weeks of fat diet (*n* = 14/group). **C** plasma cholesterol levels at sacrifice after 8 weeks of fat diet (*n* = 14/group). **D** representative photomicrographs and quantitative analysis of atherosclerotic lesions on the thoraco-abdominal aortas from chimeric female *Ldlr*[-/-]*Card9*[+/+] and *Ldlr*[-/-]*Card9*[-/-] mice after 8 weeks of fat diet (*n* = 7 *Ldlr*[-/-]*Card9*[+/+] and *n* = 6 *Ldlr*[-/-]*Card9*[-/-]); Scale bar 1 mm. **E** representative photomicrographs and quantitative analysis of atherosclerotic lesions in the aortic sinus of chimeric *Ldlr*[-/-]*Card9*[+/+] and *Ldlr*[-/-]*Card9*[-/-] mice after 8 weeks of fat diet (2 experiments pooled, *n* = 14/group); Scale bar 200 µm. **F** representative

photomicrographs and quantitative analysis of macrophage accumulation (MOMA staining, red) in atherosclerotic lesions of chimeric *Ldlr*[-/-]*Card9*[+/+] and *Ldlr*[-/-]*Card9*[-/-] mice after 8 weeks of fat diet (2 experiments pooled, *n* = 14/group); Scale bar 100 µm. **G** Representative photomicrographs and quantitative analysis of acellular area (Masson's Trichrome) of chimeric *Ldlr*[-/-]*Card9*[+/+] and *Ldlr*[-/-]*Card9*[-/-] mice after 8 weeks of fat diet (2 experiments pooled, *n* = 14/group); Scale bar 100 µm. **H** Cytokine production (ELISA in the supernatant) by Lps/Ifnγ-stimulated splenocytes isolated from chimeric *Ldlr*[-/-]*Card9*[+/+] and *Ldlr*[-/-]*Card9*[-/-] mice (*n* = 5 *Ldlr*[-/-]*Card9*[+/+] and *n* = 6 *Ldlr*[-/-]*Card9*[-/-]); Data are presented as mean values ±SD. Two-tailed Mann–Whitney test. Source data are provided as a Source Data file.

generate athero-prone lymphocyte (T, B, NKT)-deficient *Apoe*[-/-]*Rag2*[-/-]*Card9*[-/-] mice. Eight-week old male control *Apoe*[-/-]*Rag2*[-/-]*Card9*[+/+] and *Apoe*[-/-]*Rag2*[-/-]*Card9*[-/-] mice were fed a high-fat diet for 6 weeks. *Card9*-deficiency was confirmed by qPCR in peritoneal macrophages (Fig. 3A). There was no significant difference in body weight between groups (Fig. 3B) but plasma cholesterol levels were slightly increased in *Card9*-deficient mice (+17%, Fig. 3C). *Apoe*[-/-]*Rag2*[-/-]*Card9*[-/-] mice exhibited a significant increase in atherosclerotic lesion size in the thoracoabdominal aorta (+69%, Fig. 3D) and in the aortic sinus (+50%, Fig. 3E). In addition, *Card9* deletion in *Apoe*[-/-]*Rag2*[-/-] mice induced a switch toward a more inflammatory plaque phenotype with increased macrophage accumulation (Fig. 3F) and necrotic core size (Fig. 3G). Stimulated splenocytes from *Apoe*[-/-]*Rag2*[-/-]*Card9*[-/-] mice produced less TNF-α, IL-1β, and IL-10 than those from *Apoe*[-/-]*Rag2*[-/-] *Card9*[+/+] mice (Fig. 3H). Altogether, these results suggest that the adaptive immune system was not involved in the pro-atherogenic effects of *Card9* deficiency.

### *Card9* deficiency upregulates CD36 expression and increases foam cell formation

Next, we speculated that the marked increase of acellular area in atherosclerotic plaques of *Card9*-deficient mice might be related to increased foam cell formation. To explore this hypothesis, we performed in vitro experiments investigating the uptake of oxidized LDL (ox-LDL) by BM-derived macrophages and their ability to accumulate intracellular lipids. Interestingly, foam cell formation was significantly increased in *Card9*-deficient macrophages, compared with control macrophages, after 6 and 24 h of incubation with ox-LDL (Fig. 4A, B). Total cholesterol and cholesterol ester content, after ox-LDL

exposure, were significantly increased in macrophages from *Apoe*[-/-]*Card9*[-/-] mice, confirming the intracellular cholesterol overload (Fig. 4C, D). Next, we investigated the mechanisms that could drive lipid overload in the absence of Card9. We measured a significant increase in *Abca1*, *Abcg1* and *Scarb1 mRNA* levels in macrophages from *Apoe*[-/-]*Card9*[-/-] mice exposed to ox-LDL, compared to those from *Apoe*[-/-]*Card9*[+/+] mice (Fig. 4E). Cholesterol transfer to HDL and to ApoA1 was also enhanced in *Apoe*[-/-]*Card9*[-/-] macrophages (Fig. 4F). This finding highly suggests that increased foam cell formation in *Card9*-deficient macrophages was not due to impaired cholesterol efflux. Next, we investigated the expression of scavenger receptors involved in lipid uptake. We found no difference in *Msr1 mRNA* content between groups but *Cd36 mRNA* levels were markedly increased in macrophages from *Apoe*[-/-]*Card9*[-/-] mice exposed to ox-LDL (Fig. 4G), which was confirmed at the protein level by immunofluorescent staining (Fig. 4H, I). In agreement with our in vitro experiments, *Cd36 mRNA* levels were higher in aortas from *Apoe*[-/-]*Card9*[-/-] mice, compared with control mice (Supplementary Fig. 10a), as well as CD36+ macrophage numbers (Supplementary Fig. 10b).

As intracellular lipid overload modulates cell phenotype, we investigated cytokine production by macrophages and apoptosis susceptibility. Interestingly, we observed that *Card9* deficiency led to a phenotypic switch towards a more pro-inflammatory macrophage phenotype characterized by higher secretion of IL-1β and lower secretion of IL-10 following in vitro stimulation (Fig. 4J). In addition, apoptosis was increased in macrophages from *Apoe*[-/-]*Card9*[-/-] mice exposed to ox-LDL (Fig. 4K, L and Supplementary Fig. 10c). On the same note we found that the number of TUNEL+ cells was increased in atherosclerotic plaques of *Card9*-deficient mice (Fig. 4M, N and Supplementary

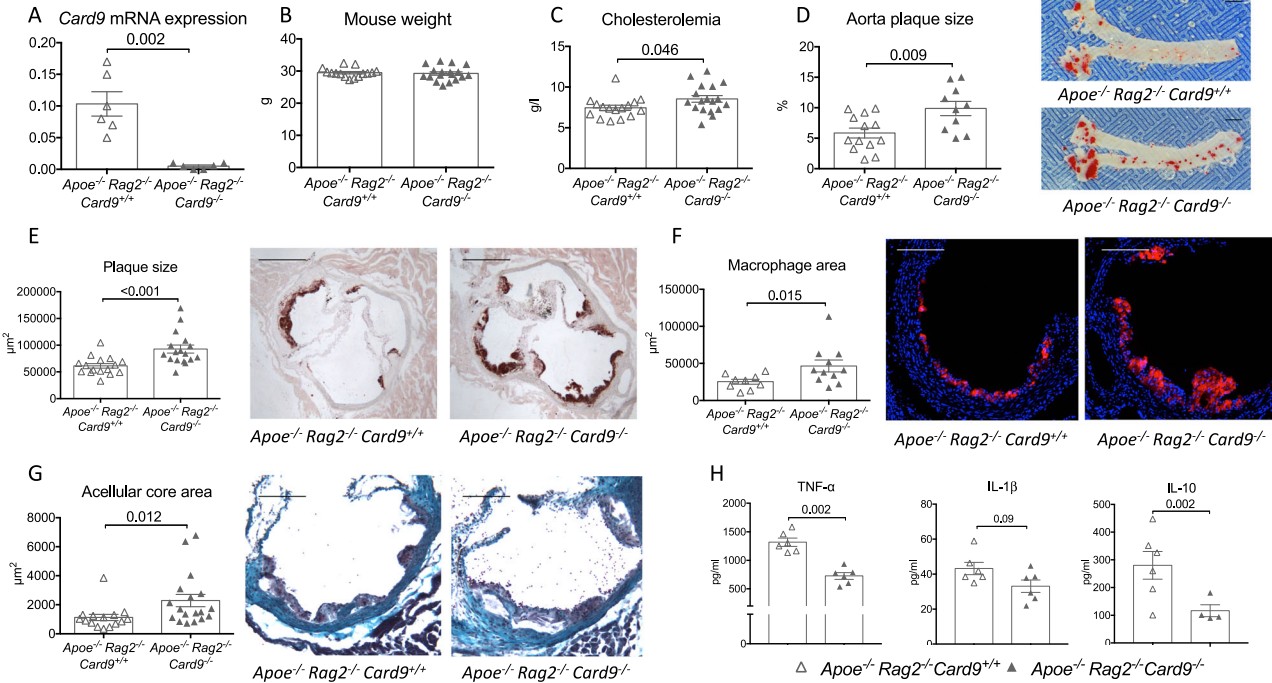

**Fig. 3 | Card9 deficiency accelerates atherosclerosis in immunodeficient *Apoe⁻/⁻Rag2⁻/⁻* mice. A** *Card9* mRNA expression by peritoneal macrophages isolated from male *Apoe⁻/⁻Rag2⁻/⁻Card9⁺/⁺* and *Apoe⁻/⁻Rag2⁻/⁻Card9⁻/⁻* mice (n = 6/group). **B** Weight at sacrifice after 6 weeks of fat diet (3 pooled experiments, n = 16–18/group). **C** plasma cholesterol levels at sacrifice after 6 weeks of fat diet (3 pooled experiments, n = 14 *Apoe⁻/⁻Rag2⁻/⁻Card9⁺/⁺* and n = 16 *Apoe⁻/⁻Rag2⁻/⁻Card9⁻/⁻*). **D** representative photomicrographs and quantitative analysis of atherosclerotic lesions on the thoraco-abdominal aorta of of *Apoe⁻/⁻Rag2⁻/⁻Card9⁺/⁺* and *Apoe⁻/⁻Rag2⁻/⁻Card9⁻/⁻* mice after 6 weeks of fat diet (n = 6 *Apoe⁻/⁻Rag2⁻/⁻Card9⁺/⁺* and n = 7 *Apoe⁻/⁻Rag2⁻/⁻Card9⁻/⁻*); **E** representative photomicrographs and quantitative analysis of atherosclerotic lesions in the aortic sinus of *Apoe⁻/⁻Rag2⁻/⁻Card9⁺/⁺* and *Apoe⁻/⁻Rag2⁻/⁻Card9⁻/⁻* mice after 6 weeks of fat diet (3 experiments pooled, n = 14 *Apoe⁻/⁻Rag2⁻/⁻Card9⁺/⁺* and n = 16 *Apoe⁻/⁻Rag2⁻/⁻Card9⁻/⁻*); Scale bar 200 µm. **F** representative photomicrographs and quantitative analysis of macrophage accumulation (MOMA staining, red) in atherosclerotic lesions of *Apoe⁻/⁻Rag2⁻/⁻Card9⁺/⁺* and *Apoe⁻/⁻Rag2⁻/⁻Card9⁻/⁻* mice after 6 weeks of fat diet (3 experiments pooled, n = 14 *Apoe⁻/⁻Rag2⁻/⁻Card9⁺/⁺* and n = 16 *Apoe⁻/⁻Rag2⁻/⁻Card9⁻/⁻*); Scale bar 100 µm. **G** Representative photomicrographs and quantitative analysis of acellular area (Masson's Trichrome) of *Apoe⁻/⁻Rag2⁻/⁻Card9⁺/⁺* and *Apoe⁻/⁻Rag2⁻/⁻Card9⁻/⁻* mice after 6 weeks of fat diet (3 experiments pooled, n = 14 *Apoe⁻/⁻Rag2⁻/⁻Card9⁺/⁺* and n = 16 *Apoe⁻/⁻Rag2⁻/⁻Card9⁻/⁻*); Scale bar 100 µm. **H** Cytokine production (ELISA) by Lps/Ifnγ-stimulated splenocytes isolated from *Apoe⁻/⁻Rag2⁻/⁻Card9⁺/⁺* and *Apoe⁻/⁻Rag2⁻/⁻Card9⁻/⁻* mice (n = 6/group). Data are presented as mean values ±SD. Two-tailed Mann–Whitney test. Source data are provided as a Source Data file.

Fig. 10d), which might also account for the presence of large necrotic core in plaques of *Apoe⁻/⁻Card9⁻/⁻* mice (Figs. 1G, 2G, and 3G).

## Acceleration of atherosclerosis in *Card9* deficiency is due to impaired autophagy

The phenotype of *Card9*-deficient mice, in terms of size and composition of atherosclerotic plaques, as well as macrophage apoptosis susceptibility, resembled that of *Atg5^fl/fl LysM^Cre+/− Ldlr⁻/⁻* mice that are characterized by impaired autophagy flux[24]. Interestingly, CD36, which was up-regulated in *Card9*-deficient mice, has been reported to participate in the regulation of autophagy in hepatocytes through AMPK downstream pathway[25]. Therefore, we next focused on the potential role of autophagy, a compensatory survival mechanism involved in atherosclerosis[26], in the context of *Card9* deficiency both in vitro and in vivo. Macrophages isolated from *Apoe⁻/⁻Card9⁺/⁺* and *Apoe⁻/⁻Card9⁻/⁻* mice were cultured and stressed to activate autophagy. Interestingly, we found a significant decrease in AMPK phosphorylation in *Card9*-deficient macrophages (Fig. 5A) but no difference in CHOP levels as well as Beclin-1 and LKB-1 phosphorylation (Supplementary Fig. 11). In addition, we found higher p62 protein content in macrophages from *Apoe⁻/⁻Card9⁻/⁻* mice compared to *Apoe⁻/⁻Card9⁺/⁺* mice (Fig. 5A). Confocal analysis of immunofluorescence staining of the p62 protein confirmed it was increased in the cytoplasm of unstimulated *Card9*-deficient macrophages, and much more in the presence of oxLDL (Fig. 5B). The p62 protein was co-localized with large inclusion bodies. LC3B dot size was significantly larger in the cytoplasm of *Apoe⁻/⁻Card9⁻/⁻* macrophages, with specific aberrant

colocalization with these large p62+ inclusion bodies (Fig. 5B). P62+ inclusions bodies have been previously described in *Atg5*-null macrophages[27], which strongly supports impaired autophagy in the absence of Card9. The accumulation of p62 in *Card9*-deficient macrophages was confirmed in vivo, as revealed by the higher number of p62+ MOMA+ macrophages in atherosclerotic plaques of *Apoe⁻/⁻Card9⁻/⁻* mice in reference to plaques of *Apoe⁻/⁻Card9⁺/⁺* mice (Fig. 5C). Next, we evaluated a pharmacological approach to restore autophagy treated with Metformin, a well-known activator of autophagy through AMPK stimulation[28]. In vitro, metformin treatment abolished the accumulation of p62+ protein in the *Card9*-deficient macrophages and the formation of inclusion bodies (Fig. 5D). In vivo, *Apoe⁻/⁻Card9⁺/⁺* and *Apoe⁻/⁻Card9⁻/⁻* mice, under a high-fat diet, were treated or not with metformin during 6 weeks[28]. At sacrifice, there was no significant difference in body weight and in plasma cholesterol levels between *Apoe⁻/⁻Card9⁺/⁺* and *Apoe⁻/⁻Card9⁻/⁻* groups treated with metformin (Supplementary Fig. 12a, b). Metformin treatment abolished the acceleration of atherosclerosis observed in *Card9* deficiency, with no difference in plaque size (Fig. 5E), plaque composition (Fig. 5F, G) and P62 accumulation (Fig. 5H) between the 2 treated groups.

Next, we used an addition complementary pharmacological approach to activate autophagy with rapamycin, an inhibitor of both mTORC1 and mTORC2[29]. In vitro, rapamycin treatment restored autophagy flux, as shown by strong reduction of p62 protein accumulation in the cytoplasm of *Card9*-deficient macrophages and an almost disparition of p62+ inclusion bodies (Fig. 6A, B). In addition, rapamycin treatment abolished intracellular lipid overload (Fig. 6C),

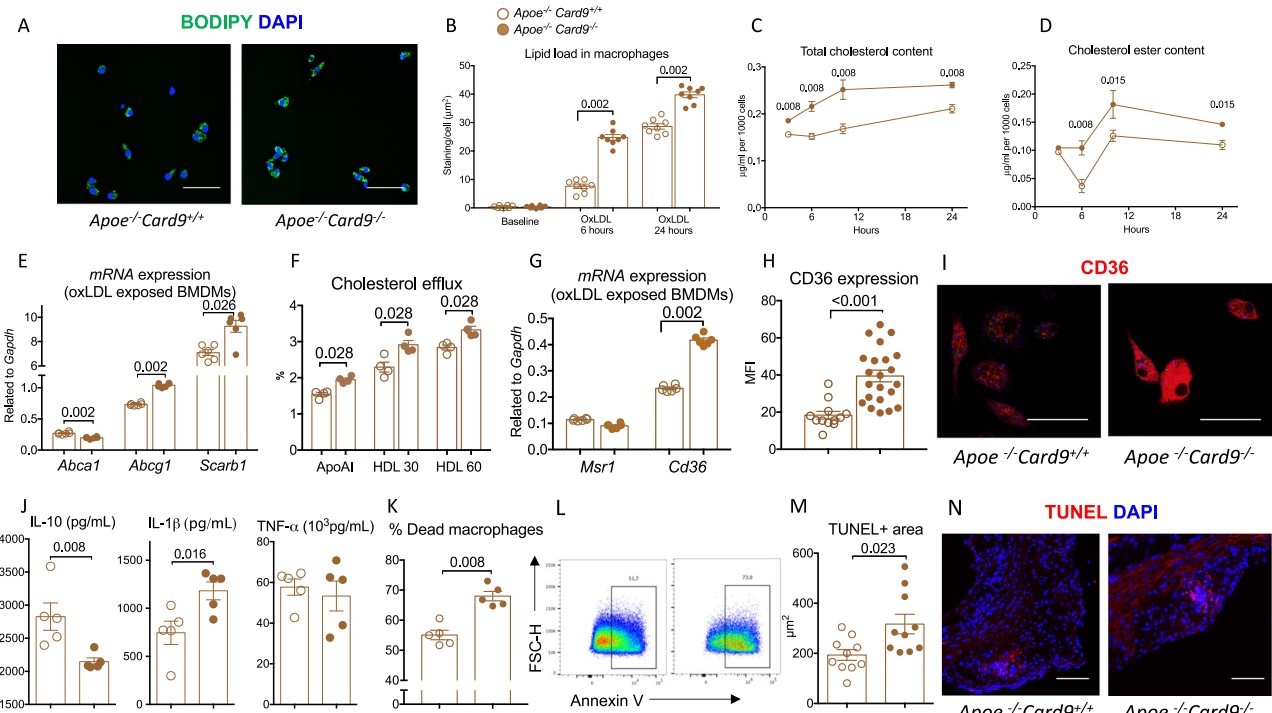

**Fig. 4 | *Card9* deficiency increased CD36 expression in macrophages and promotes lipid uptake and foam cell formation. A** Representative photomicrographs and quantitative analysis (**B**) of Bodipy+ foam cells after incubation of BMDMs from *Apoe⁻/⁻Card9⁺/⁺* and *Apoe⁻/⁻Card9⁻/⁻* mice with oxLDL during 6 and 24 h (2 pooled experiments, *n* = 8/group/timepoints), Scale bar 10 μm. **C** quantification of intracellular cholesterol content of BMDMs from *Apoe⁻/⁻Card9⁺/⁺* and *Apoe⁻/⁻Card9⁻/⁻* mice after exposure to ox-LDL (*n* = 6/group/timepoint). **D** Quantification of intracellular cholesterol ester of BMDMs from *Apoe⁻/⁻Card9⁺/⁺* and *Apoe⁻/⁻Card9⁻/⁻* mice after exposure to ox-LDL at 3, 6, 12, and 24 h (*n* = 6/group/timepoint). **E** Quantification of *Abca1* and *Abcg1* and *Scarb1* mRNA expression in BMDMs from *Apoe⁻/⁻Card9⁺/⁺* and *Apoe⁻/⁻Card9⁻/⁻* mice after stimulation with oxLDL (*n* = 6/group). **F** Quantification of cholesterol efflux in presence of ApoAI or HDL (*n* = 4/group). **G** Quantification of *Mrs1 and Cd36* mRNAs in BMDMs from *Apoe⁻/⁻Card9⁺/⁺* and *Apoe⁻/⁻Card9⁻/⁻* mice after stimulation with oxLDL (*n* = 6/group). **H, I** representative immunostaining and quantification of Cd36 expression by BMDMs from *Apoe⁻/⁻Card9⁺/⁺* and *Apoe⁻/⁻Card9⁻/⁻* mice after 24-h stimulation with ox-LDL (*n* = 12 *Apoe⁻/⁻Card9⁺/⁺* and *n* = 22 *Apoe⁻/⁻Card9⁻/⁻*), Scale bar 10 μm. **J** cytokine production by BMDMs from *Apoe⁻/⁻Card9⁺/⁺* and *Apoe⁻/⁻Card9⁻/⁻* mice after stimulation (ELISA in the supernatant, *n* = 5/group) and oxLDL exposure. **K, L** Flow cytometry quantification of AnnexinV+ BMDMs from *Apoe⁻/⁻Card9⁺/⁺* and *Apoe⁻/⁻Card9⁻/⁻* mice after 24-h stimulation with oxLDL (100 μmol/l; *n* = 5/group). **M, N** representative photomicrographs and quantification of TUNEL+ cells in plaques from *Apoe⁻/⁻Card9⁺/⁺* and *Apoe⁻/⁻Card9⁻/⁻* mice (*n* = 10/group). Scale bar 20 μm. Data are presented as mean values ±SD. Two-tailed Mann–Whitney test. Source data are provided as a Source Data file.

and attenuated cell death susceptibility observed in *Card9*-deficient macrophages (Fig. 6D). To further validate the physiopathological relevance of our findings, *Apoe⁻/⁻Card9⁺/⁺* and *Apoe⁻/⁻Card9⁻/⁻* mice were fed a high-fat diet for 6 weeks and received daily Rapamycin treatment. There was no difference in body weight and in plasma cholesterol levels between *Apoe⁻/⁻Card9⁺/⁺* and *Apoe⁻/⁻Card9⁻/⁻* groups treated with rapamycin (Supplementary Fig. 12c, d). Interestingly, rapamycin treatment abolished the pro-atherogenic effect of *Card9* deficiency. Histological analysis revealed a significant decrease in atherosclerosis in the aortic sinus of rapamycin-treated *Apoe⁻/⁻Card9⁻/⁻* mice, compared to rapamycin-treated *Apoe⁻/⁻Card9⁺/⁺* mice (Fig. 6D). In addition, macrophage content and necrotic core size were also significantly reduced in atherosclerotic plaques of rapamycin-treated *Apoe⁻/⁻Card9⁻/⁻* mice (Fig. 6F, G), as well as the accumulation of p62 (Fig. 6H).

### *Card9* deficiency-mediated acceleration of atherosclerosis is dependent on CD36

To address the direct role of CD36, we generated *Cd36*-deficient *Card9⁺/⁺* and *Card9⁻/⁻* mice. Following in vitro challenge, AMPK phosphorylation (Fig. 7A), p62 accumulation (Fig. 7B), lipid uptake (Fig. 7C and supplementary Fig. 13a, b) and cell death susceptibility (Fig. 7D) were not different between macrophages from *Cd36⁻/⁻Card9⁺/⁺* and *Cd36⁻/⁻Card9⁻/⁻* mice. Furthermore, increased production of IL-1β observed in *Card9⁻/⁻* macrophages was also abolished in the absence of *Cd36* (Fig. 7E).

*Apoe⁻/⁻Card9⁺/⁺* and *Apoe⁻/⁻Card9⁻/⁻* mice after stimulation with oxLDL (*n* = 6/group).

To investigate the effect of *Card9* deficiency on atherosclerosis in the absence of CD36, we performed BM transplantation experiments using either *Cd36⁻/⁻Card9⁺/⁺* or *Cd36⁻/⁻Card9⁻/⁻* littermate BM to repopulate lethally irradiated male *Ldlr⁻/⁻* mice. After 4 weeks of recovery and additional 8 weeks of high-fat diet, animals were sacrified. We confirmed that *Cd36* and *Card9* gene expression was almost abolished in the spleen (Fig. 7F, G) and in peritoneal macrophages of chimeric *Ldlr⁻/⁻* (Supplementary Fig. 13c). Cholesterol levels were not different between both groups (Fig. 7H). Atherosclerosis plaque size and composition were no longer different between *Ldlr⁻/⁻Cd36⁻/⁻Card9⁺/⁺* and *Ldlr⁻/⁻Cd36⁻/⁻Card9⁻/⁻* chimeric groups (Fig. 7I, K), indicating that increased expression of CD36 was directly responsible for increased atherosclerosis in *Card9*-deficient animals.

### CARD9-related pathways in human

In order to evaluate the clinical relevance of our experimental results, we compared the transcriptomic profile of blood monocytes from *CARD9*-deficient patients (*n* = 3, 2 males and 1 female) with those of controls (*n* = 4, 2 males and 2 females) (Fig. 8A and Supplementary data 1a, b). Genetic investigations in *CARD9*-deficient patients identified a homozygous c.865C > T in exon 6 of CARD9 leading to a premature stop codon at position 289 (p.Q289*) (*n* = 2, 1 male and 1 female) and a homozygous c.52 C > T missense mutation in exon 2 of CARD9, resulting in the replacement of the arginine in position 18 with a tryptophan residue (p.R18W) (*n* = 1 female)[30–32]. Differential analysis

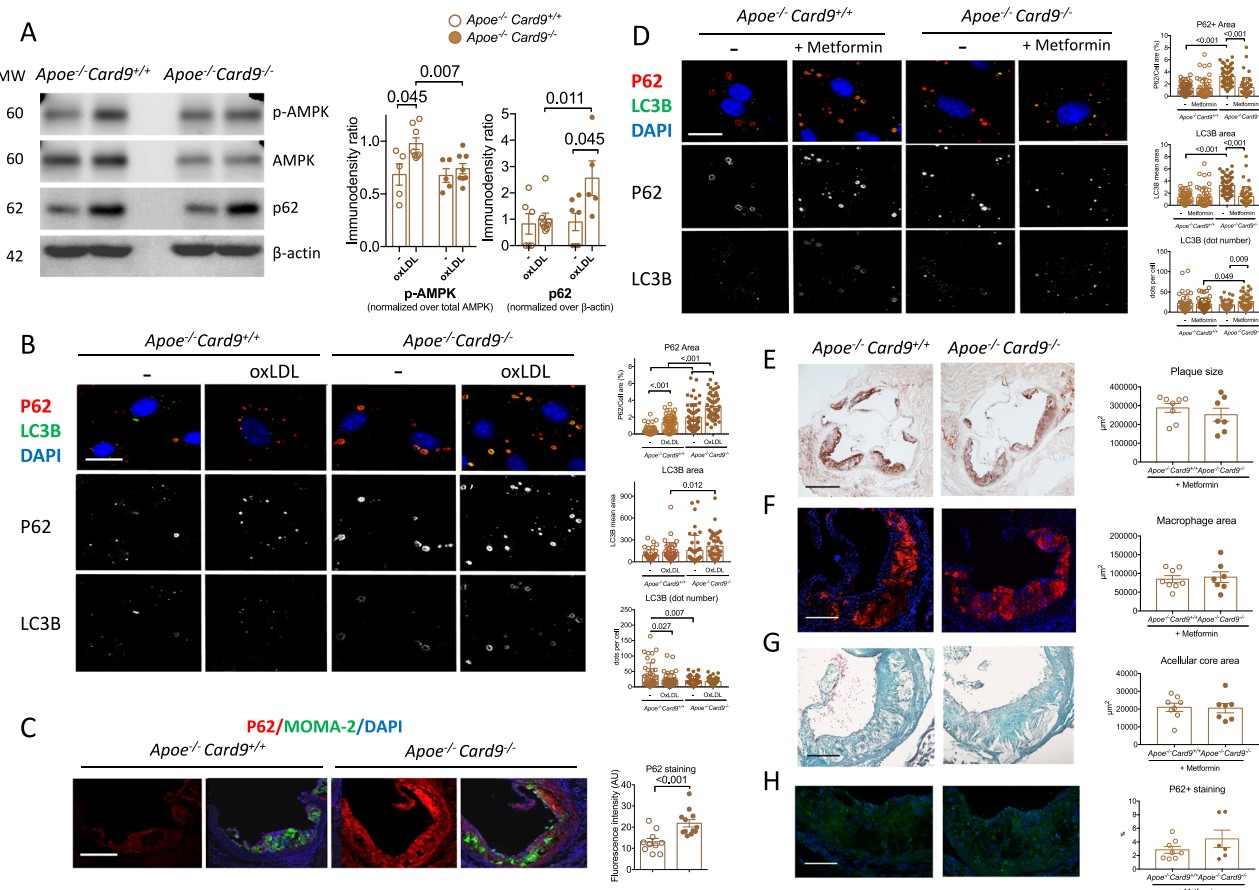

**Fig. 5 | Pro-atherogenic effects of *Card9* deficiency are mediated by impaired autophagy. A** Quantitation of AMPK phosphorylation and p62 content in macrophages from from *Apoe^-/- Card9^+/+* (*n* = 5) and *Apoe^-/- Card9^-/-* mice (*n* = 8) after exposure to oxLDL (western blot). **B** representative photomicrographs and quantitative analysis and of p62 (Red), LC3B (Green) content in macrophages from *Apoe^-/- Card9^+/+* and *Apoe^-/- Card9^-/-* mice at baseline and after exposure to oxLDL (*n* = 48 for p62 and *n* = 40 for LC3B). **C** Quantitative analysis and representative photomicrographs of p62 accumulation (Red) in MOMA-2+ macrophages (Green) in atherosclerotic lesions of *Apoe^-/- Card9^+/+* (*n* = 10) and *Apoe^-/- Card9^-/-* mice (*n* = 12) after 6 weeks of fat diet (2 experiments pooled); Scale bar 200 µm. **D** representative photomicrographs and quantitative analysis and of p62 (Red), LC3B (Green) content in ox-LDL exposed macrophages from *Apoe^-/- Card9^+/+* and *Apoe^-/- Card9^-/-* mice treated or not with metformin (*n* = 48 for p62 and *n* = 40 for LC3B). **E–H** characterization of atherosclerotic lesions in the aortic sinus of male

*Apoe^-/- Card9^+/+* (*n* = 8) and *Apoe^-/- Card9^-/-* mice (*n* = 7) after 6 weeks of fat diet and treated by metformin. **E** representative photomicrographs and quantitative analysis of atherosclerotic lesions in the aortic sinus of *Apoe^-/- Card9^+/+* and *Apoe^-/- Card9^-/-* mice after 6 weeks of fat diet and treated by metformin; Scale bar 200 µm. **F** representative photomicrographs and quantitative analysis of macrophage accumulation (MOMA staining, red) in atherosclerotic lesions of *Apoe^-/- Card9^+/+* and *Apoe^-/- Card9^-/-* mice after 6 weeks of fat diet and treated by metformin; Scale bar 100 µm. **G** representative photomicrographs and quantitative analysis of acellular area (Masson's Trichrome) of *Apoe^-/- Card9^+/+* and *Apoe^-/- Card9^-/-* mice after 6 weeks of fat diet and treated by metformin; Scale bar 100 µm. **H** representative photomicrographs and quantitative analysis of p62 content (Green) in plaques of *Apoe^-/- Card9^+/+* and *Apoe^-/- Card9^-/-* mice after 6 weeks of fat diet and treated by metformin; Scale bar 100 µm. Data are presented as mean values ±SD. Two-tailed Mann–Whitney test. Source data are provided as a Source Data file.

of these RNA-Seq revealed 256 differentially expressed genes (Fig. 8B): 211 were up-regulated and 45 were down-regulated in *CARD9*-deficient patients. Up-regulated genes included inflammatory cytokines, such as IL-1β [log2(FC) = 2.2, adjusted *p*-value = 0.035] and IL-6 [log2(FC) = 4.3, adjusted *p*-value = 1.52e-05] (Fig. 8C). In gene set enrichment analysis (GSEA), several pathways previously identified in murine models had significant enriched score: apoptosis (34 pathways, 330 genes), autophagy (1 pathway, 1 gene), atherosclerosis (2 pathways, 69 genes), NF-κB signaling (1 pathway, 25 genes) and TNFα signaling pathway (1 pathway, 20 genes) (Fig. 8D and detailed selected pathways and core enrichment genes in Supplementary data 2).

Finally, we examined CARD9 expression in atherosclerotic plaques from human carotid arteries. CARD9 was not detected in normal aorta (Supplementary Fig. 14a). However, CARD9 was detected in fatty streak lesions (Fig. 8D) and in lipid-rich areas surrounding the necrotic core of advanced atherosclerotic plaques (Fig. 8E). CARD9 expression was higher in atheromatous plaques than in fibrous lesions (Supplementary Fig. 14b). Fluorescent staining confirmed that CARD9

expression was mostly confined to CD68+ intimal macrophages (Fig. 8D, E), and analysis of previously published single-cell RNA-seq[33] revealed *CARD9* mRNA expression specifically in macrophages from atherosclerotic human coronary arteries (Fig. 8F and Supplementary Fig. 15a–d). In an integrated single-cell analysis of human mononuclear phagocytes in atherosclerosis[34], *CARD9* was detected in all the major plaque macrophage subsets (Supplementary Fig. 15e–g).

## Discussion

Using several in vitro and in vivo complementary approaches, we demonstated that *Card9* deficiency significantly accelerated atherosclerosis in mice throughout the aorta (aortic sinus, ascending and descending aorta) and induced a more inflammatory plaque phenotype, characterized by increased macrophage infiltration and necrotic core size. *Card9*-deficient macrophages are characterized by increased pro-inflammatory cytokine release, enhanced lipid uptake and elevated cell death susceptibility (Supplementary Fig. 16). Such proatherogenic phenotype resulted from CD36-mediated impaired

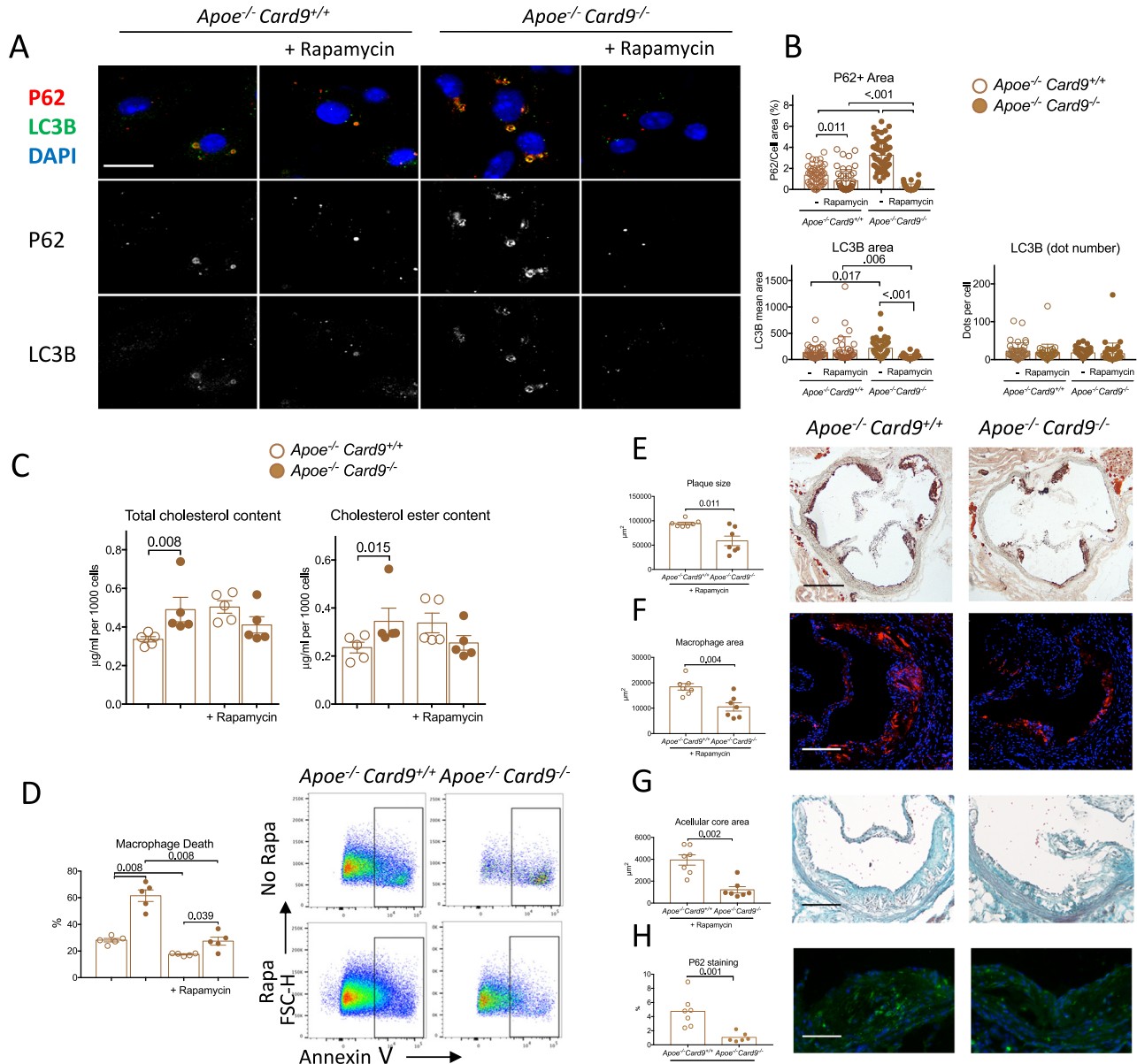

**Fig. 6 | Protective effects of rapamycin in vitro and in vivo. A**, **B** representative photomicrographs and quantitative analysis of p62 (Red), LC3B (Green) content in oxLDL-exposed macrophages from male *Apoe-/-Card9+/+* and *Apoe-/-Card9-/-* mice treated or not with rapamycin (200 mMol) (*n* = 48 for p62 and *n* = 40 for LC3B). **C** quantification of intracellular cholesterol (total and ester) of BMDMs from *Apoe-/-Card9+/+* and *Apoe-/-Card9-/-* mice after exposure to oxLDL with or without rapamycin (200 mMol) (*n* = 5/group). **D** Flow cytometry quantification of necrosis (7AAD+ Annexin V+) susceptibility of macrophages from *Apoe-/-Card9+/+* and *Apoe-/-Card9-/-* mice after 24-h stimulation with high concentration of oxLDL (100 µmol/l) without and with rapamycin (200 mMol; *n* = 5/group). **E** representative photomicrographs and quantitative analysis of atherosclerotic lesions in the aortic sinus of *Apoe-/-Card9+/+* and *Apoe-/-Card9-/-* mice after 6 weeks of fat diet and treated by rapamycin (*n* = 7/group); Scale bar 200 µm. **F** representative photomicrographs and quantitative analysis of macrophage accumulation (MOMA staining, red) in atherosclerotic lesions of *Apoe-/-Card9+/+* and *Apoe-/-Card9-/-* mice after 6 weeks of fat diet and treated by rapamycin (*n* = 7/group; Scale bar 100 µm). **G** representative photomicrographs and quantitative analysis of acellular area (Masson's Trichrome) of *Apoe-/-Card9+/+* and *Apoe-/-Card9-/-* mice after 6 weeks of fat diet and treated by rapamycin (*n* = 7/group; Scale bar 100 µm). **H** representative photomicrographs and quantitative analysis of p62 content (green) in plaques of *Apoe-/-Card9+/+* and *Apoe-/-Card9-/-* mice after 6 weeks of fat diet and treated by rapamycin (*n* = 7/group; Scale bar 100 µm.). Data are presented as mean values ±SD. Two-tailed Mann–Whitney test. Source data are provided as a Source Data file.

autophagy, since this was reversed by *Cd36* gene deletion and by autophagy-inducing treatments with rapamycin and metformin. Importantly, the human relevance of our findings in mice was confirmed using transcriptomic analysis of monocytes isolated from extremely rare *CARD9*-deficient patients and ScRNA seq analysis in human atherosclerotic plaques.

The role of CARD9 in atherosclerosis still remains debated. In two recent studies, one reported increased lesion size in chimeric *Ldlr-/-Card9-/-* mice[16], whereas the other found that deletion of haematopoietic Card9 did not affect the atherosclerosis in chimeric *Ldlr-/-* mice under hyperglycaemic conditions[17]. In our study, we provided strong evidence that *Card9* deficiency promoted atherosclerosis. We confirmed this finding in two different atheroprone models, *Apoe-/-* mice and chimeric *Ldlr-/-* mice under high-fat diet. The pro-atherogenic role of *Card9* deficiency was not dependent on gender as we observed similar accelerated vascular disease in both male *Apoe-/-* and female chimeric *Ldlr-/-* mice with *Card9* deficiency. The acceleration of atherosclerosis, which is known to be an inflammatory disease,

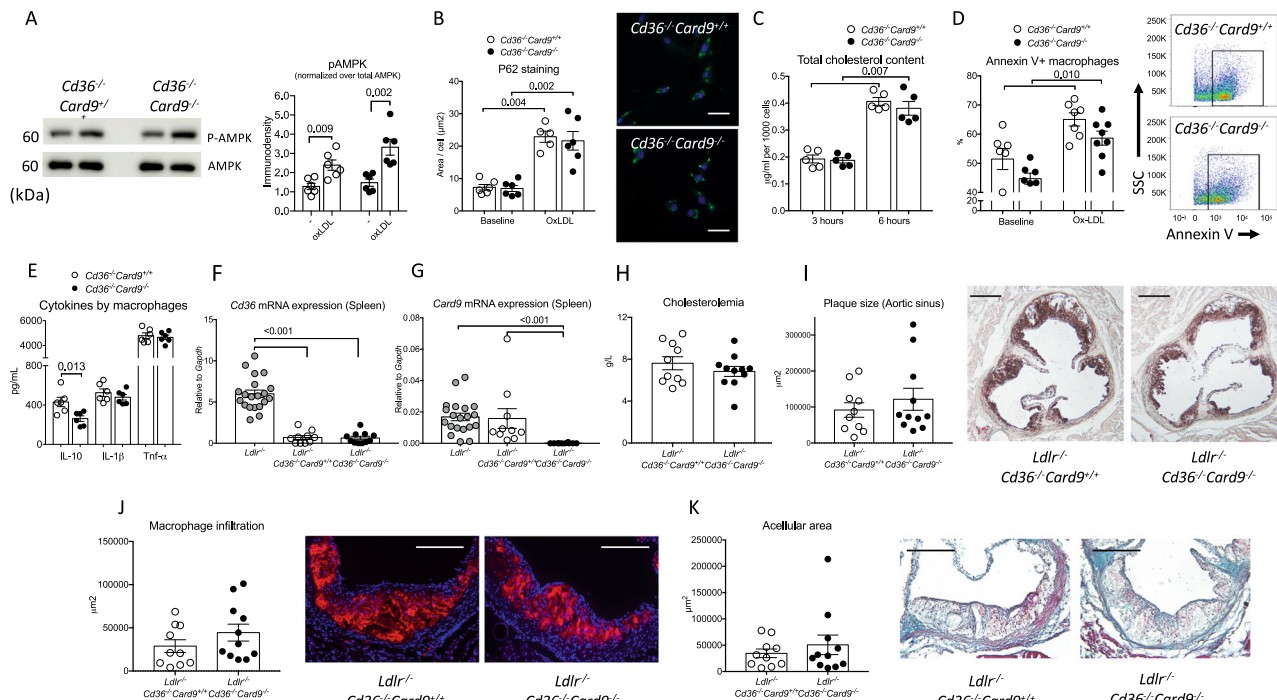

**Fig. 7 | Pro-atherogenic effect of hematopoietic *Card9* deficiency was abolished in the absence of Cd36. A** representative blots and quantification of AMPK phosphorylation on BM-derived macrophages from *Cd36⁻/⁻Card9⁺/⁺* and *Cd36⁻/⁻Card9⁻/⁻* macrophages after exposure to oxLDL (*n* = 6/group/condition). **B** Quantitative analysis and representative photomicrographs of P62 accumulation in macrophages from *Cd36⁻/⁻Card9⁺/⁺* and *Cd36⁻/⁻Card9⁻/⁻* macrophages after 8-h exposure to oxLDL, Scale bar 100 μm (*n* = 6/group/condition). **C** quantification of intracellular Cholesterol of BM-derived macrophages from *Cd36⁻/⁻Card9⁺/⁺* and *Cd36⁻/⁻Card9⁻/⁻* mice after 3 and 6 h exposure to ox-LDL (*n* = 5/group/timepoint). **D** Flow cytometry quantification of Annexin V+ apoptotic BM-derived macrophages from *Cd36⁻/⁻Card9⁺/⁺* and *Cd36⁻/⁻Card9⁻/⁻* mice after 24-h stimulation with high concentration of oxLDL (50 μmol/l; Baseline *n* = 6/group, OxLDL *n* = 7 *Cd36⁻/⁻Card9⁺/⁺* and *n* = 8 *Cd36⁻/⁻Card9⁻/⁻*). **E** Cytokine production by BMDMs from *Cd36⁻/⁻Card9⁺/⁺* and *Cd36⁻/⁻Card9⁻/⁻* mice after stimulation (ELISA in the supernatant, *n* = 6/group). **F** *Cd36* mRNA expression in the spleen from *Ldlr⁻/⁻* mice and chimeric

*Ldlr⁻/⁻Cd36⁻/⁻Card9⁺/⁺* and *Ldlr⁻/⁻Cd36⁻/⁻Card9⁻/⁻* mice (*n* = 20/10/10). **G** *Card9* mRNA expression in the spleen from *Ldlr⁻/⁻* mice and chimeric *Ldlr⁻/⁻Cd36⁻/⁻Card9⁺/⁺* and *Ldlr⁻/⁻Cd36⁻/⁻Card9⁻/⁻* mice (*n* = 20/10/10). **H–K** characterization of chimeric male *Ldlr⁻/⁻Cd36⁻/⁻Card9⁺/⁺* (*n* = 10) and *Ldlr⁻/⁻Cd36⁻/⁻Card9⁻/⁻* mice (*n* = 11). **H** cholesterolemia in chimeric mice at sacrifice. **I** representative photomicrographs and quantitative analysis of atherosclerotic lesions in the aortic sinus of chimeric *Ldlr⁻/⁻Cd36⁻/⁻Card9⁺/⁺* and *Ldlr⁻/⁻Cd36⁻/⁻Card9⁻/⁻* mice after 8 weeks of fat diet; Scale bar 200 μm. **J** representative photomicrographs and quantitative analysis of macrophage accumulation (MOMA staining, red) in atherosclerotic lesions of chimeric *Ldlr⁻/⁻Cd36⁻/⁻Card9⁺/⁺* and *Ldlr⁻/⁻Cd36⁻/⁻Card9⁻/⁻* mice after 8 weeks of fat diet; Scale bar 100 μm. **K** Representative photomicrographs and quantitative analysis of acellular area (Masson's Trichrome) of chimeric *Ldlr⁻/⁻Cd36⁻/⁻Card9⁺/⁺* and *Ldlr⁻/⁻Cd36⁻/⁻Card9⁻/⁻* mice after 8 weeks of fat diet; Scale bar 100 μm. Data are presented as mean values ±SD. Two-tailed Mann–Whitney test. Source data are provided as a Source Data file.

in the absence of *Card9* might be counterintuitive since Card9 is a downstream adapter of fungal- and bacteria-induced activation of TLRs, as well as activation of ITAM-containing non-TLRs and Dectin-1[35,36]. Card9 engagement and Card9-Bcl-10-MALT1 complex formation lead to NF-κB transcription and subsequent secretion of pro-inflammatory cytokines[37]. Decreased spleen production of TNF-α is consistent in all murine models and in line with previous studies[35]. The effect of *Card9* deficiency on IL-1β production is more complex. We found decreased IL-1β production by stimulated mixed immune cells in the spleen, but higher production by murine *Card9⁻/⁻* macrophages as well as higher *IL-1β* transcripts in monocytes from *CARD9*-deficient patients. In the context of *Salmonella* infection, it has also been reported that CARD9 negatively regulates IL-1β by fine-tuning pro-IL-1β expression, SYK-mediated NLRP3 activation and repressing inflammasome-associated caspase-8 activity[38]. Higher production of IL-1β might be involved, at least in part, in the acceleration of atherosclerosis in *Apoe⁻/⁻Card9⁻/⁻* mice and might be due to increased CD36 expression. Liu et al. have shown that CD36 promoted the expression of NLRP3 and consecutive IL-1β production through ROS generation in ox-LDL-stimulated macrophages[39]. In our study, we found that Card9 deficiency had significant effects on the adaptive immune system and particularly on T cell polarization. We observed discrepancies in cytokine production by CD4 + T cells between male *Apoe⁻/⁻* and female chimeric *Ldlr⁻/⁻*, which could be due to gender or background

difference, Apoe having by itself immune-modulatory functions[40]. In chimeric *Ldlr⁻/⁻* mice, *Card9*-deficient CD4⁺ T cells produced less IL-17A than control CD4⁺ T cells, which is consistent with previous studies in normocholesterolemic mice[13]. However, our findings of increased atherosclerosis in immune-deficient *Apoe⁻/⁻Rag2⁻/⁻Card9⁻/⁻* mice ruled out the possibility that the acceleration of atherosclerosis in the absence of Card9 was mediated by a modulation of the adaptive immune system.

Given the marked increase in aortic atherosclerosis in the absence of hematopoietic Card9 and the colocalization between Card9+ macrophages and lipid-rich areas in both mouse and human plaques, we then focused on the role of Card9 in macrophage foam cell formation. We found a marked increase in ox-LDL uptake and lipid accumulation in *Card9⁻/⁻* macrophages. Among the receptors that govern foam cell formation in macrophages, *Card9* deletion selectively increased both Cd36 gene expression and cell surface protein levels, in vitro, as well as in vivo in plaque macrophages. The effect of *Card9* deficiency on the upregulation of CD36 expression and lipid uptake might account for the accelerated atherosclerosis in *Card9*-deficient mice. Several studies have previously reported a pro-atherogenic role of CD36 in *Apoe⁻/⁻*[41] and *Ldlr⁻/⁻* mice[42].

We reported increased apoptosis susceptibility of *Card9⁻/⁻* macrophages exposed to ox-LDL, which is consistent with the increase number of TUNEL+ cells found in plaques of *Apoe⁻/⁻Card9⁻/⁻* mice.

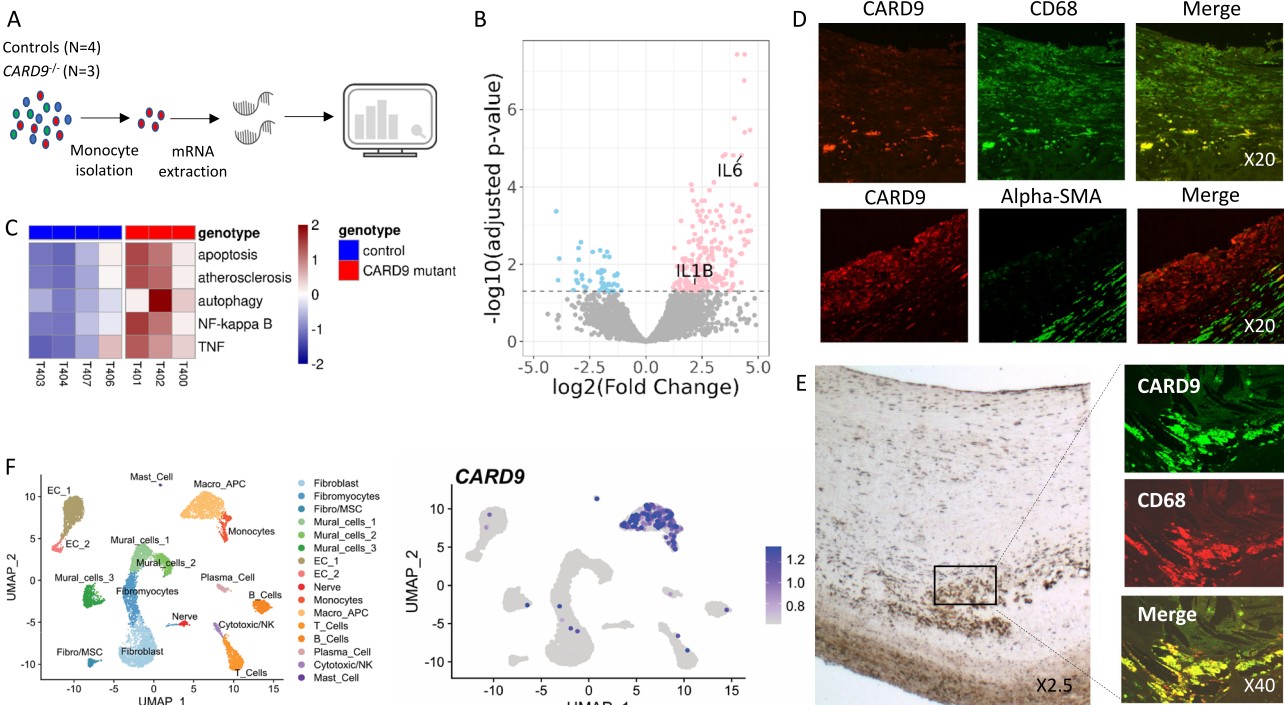

**Fig. 8 | CARD9 related pathways in human. A** protocol to obtain transcripts after isolation of monocytes from controls and *CARD9*-deficient patients. **B** Volcano-plot of the differentially expressed genes between monocytes from patients with *CARD9* mutation and control patients. Red dots represent up-regulated genes and blue dots down-regulated genes (adjusted *p*-value < 0.05). **C** Heatmap of mean expression in each patient of the leading edge genes contributing to the enrichment of indicated pathways in the GSEA. Immunofluorescent micrograph of human healthy (**D**) and atherosclerotic (**E**) carotid artery sections stained for CARD9 (green), α-actin + smooth muscle (red) or CD68 (red) showing that CARD9 was strongly expressed by cells that engulf lipids and cholesterol crystals and of giant lipid-laden foam cells but not by smooth muscle cells. Magnitude X20 (D), X2.5 (E), X40 (E) (2 Pooled experiments, *n* = 6/staining). **F** CARD9 expression in 10,934 total human atherosclerotic coronary artery cells from 4 patients (data from Wirka et al.[33]) with UMAP representation of single-cell RNA-seq gene expression data (left) and cellular lineage identification (right) where CARD9 expression in single cells projected onto the UMAP plot. (For clarity, expression cutoff have been applied and cells with detectable Card9/CARD9 transcripts were brought to the front of the plot).

Apoptosis susceptibility of *Card9*-deficient macrophages might be due to intracellular lipid overload. In the context of atherosclerosis, other mechanisms, including autophagy, could account for enhanced apoptosis susceptibility of macrophages in the absence of Card9. Notably, we found that autophagy was impaired in cultured *Card9*-deficient macrophages, as well as in atherosclerotic plaques of *Apoe*[-/-]*Card9*[-/-] mice, as shown by the accumulation of p62 accumulation. Confocal analysis revealed that the p62 protein accumulated in large inclusion bodies with LC3B. These inclusion bodies that are p62-enriched polyubiquitinated protein aggregates, have been previously described in the context of impaired autophagy[27]. These observations highly suggested that Card9 deficiency impaired autophagy flux, but the exact molecular mechanisms requires further investigation. Ohman et al. previously suggested a link between the Card9 signaling pathway and autophagy, and found that Curdlan, an activator of the Dectin-1 receptor, induced LC3I conversion into LC3II in cultured macrophage[43]. More recently, Rubicon, a Beclin-1-binding partner, was identified as a physiological feedback inhibitor of Card9-BCL10-MALT1-mediated PRR signaling[44]. Impaired autophagy has also been shown to promote atherosclerosis. Autophagy blockade in *LysMCre*[+]*Atg5l*[lox/lox]*Ldlr*[-/-] mice led to increased lesion size and larger necrotic core[24], which phenocopies the genetic loss of *Card9* in *Apoe*[-/-] mice. Our study showed that impaired autophagy in *Card9*-deficient macrophages was mediated, at least in part, by AMPK blockade. We focused on AMPK because CD36 is known to inhibit AMPK phosphorylation[25,45] and also because AMPK is an upstream regulator of autophagy through several mechanisms, including Ulk1 activation[46] and mTORC1 inhibition[47]. In *Card9*-deficient macrophages, oxLDL-induced AMPK phosphorylation was abolished and P62 protein

accumulated in cell cytoplasm. Interestingly, AMPK phosphorylation and autophagy were restored in *Cd36*[-/-]*Card9*[-/-] cells, supporting a critical role of CD36 in the regulation of autophagy and subsequently atherosclerotic plaque development. Rapamycin and metformin, two pharmacological activator of autophagy, respectively through mTOR blockade and AMPK activation[29], restored in vitro autophagy flux in *Card9*-deficient macrophages, with a strong reduction of p62+ inclusion bodies formation. In vivo, both treatment abolished the pro-atherogenic effect of *Card9* deficiency and reduced p62 accumulation in atherosclerotic lesions. As rapamycin and metformin have several molecular targets, it cannot be ruled out that some of their anti-atherogenic effects were independent of their effect on autophagy.

Experiments performed in *Cd36*[-/-] animals confirmed that CD36 upregulation was involved in the acceleration of atherosclerosis in *Card9*[-/-] animals. However, how the two functions of CD36 (lipid uptake *versus* autophagy) contribute to the pro-atherogenic impact of Card9 deficiency remains unknown. Addressing this issue is very challenging because autophagy, by itself, can regulate Cd36 levels[48]. Finally, in our study, cholesterol efflux was not impaired in *Card9*-deficient macrophages suggesting that autophagy-independent mechanisms such as neutral lipolysis[49,50] were activated in *Card9*[-/-] macrophages to limit lipid overload.

Finally, given the marked differences in the profiles of gut microbiota between *Apoe*[-/-] and *Ldlr*[-/-] models, but the very similar pro-atherogenic effects of *Card9* deficiency in these two murine models of atherosclerosis, it is unlikely that these vascular effects were due to Card9-associated dysbiosis. However, in the absence of microbiota transfer experiments, a contribution of gut dysbiosis in the vascular phenotype cannot be definitively ruled out.

Altogether, our studies identify CARD9 as a major protective pathway in the development and complications of atherosclerosis. Pro-apoptotic and pro-atherogenic effects of *Card9* deficiency are mediated by CD36-dependent defective autophagy that can be reversed by rapamycin and metformin.

## Methods

### Animals

Experiments were conducted according to the guidelines formulated by the European Community for experimental animal use (L358-86/609EEC) and were approved by the Ethical Committee of INSERM and the French Ministry of Agriculture (agreement A75-15-32). Mouse breeding occurred in our animal facility in accordance with local recommendations. Animals were provided with food and water *ad libitum*. All the animals were maintained under identical standard conditions (housing, regular care and high-fat diet). Mice were maintained in isolated ventilated cages under specific pathogenfree conditions. Before euthanasia by cervical dislocation, animals were anesthetized with isoflurane (3% in oxygen).

*Card9⁻/⁻* and *Rag2⁻/⁻Card9⁻/⁻* mice (null for the *Card9* gene), have been generated by Dr. Harry Sokol (Centre de Recherche Saint-Antoine UMR_S 938, Paris, France)[13]. They were backcrossed for over 10 generations into a C57BL/6 J background. *Apoe⁻/⁻Card9⁻/⁻* and *Apoe⁻/⁻Rag2⁻/⁻Card9⁻/⁻* mice have been generated in our lab and males were used for atherosclerosis studies. *Cd36⁻/⁻* mice (null for the *Cd36* gene) *C57bl/6 Cd36⁻/⁻* mice were generated in Dr. Roy Silverstein's laboratory (Febbraio et al. JBC 1999) and were crossed with *Card9⁻/⁻* mice to generate *Card9⁻/⁻Card36⁻/⁻* mice. Ten-week old female C57BL/6 J *Ldlr⁻/⁻* mice were subjected to medullar aplasia by lethal whole-body irradiation (9.5 grays). The mice were repopulated with an intravenous injection of bone marrow (BM) cells isolated from femurs and tibias of sex-matched C57BL/6 J *Card9⁻/⁻* mice or *Card9⁺/⁺* littermates. After 4 weeks of recovery, mice were fed a pro-atherogenic high-fat diet containing 15% fat, 1.25% cholesterol, and 0% cholate for 8 weeks. During the first 14 days following BM cell transfer, animals were administered an antibiotic in their drinking water (Baytril®, Enrofloxacine). Ten-week old male C57BL/6J *Ldlr⁻/⁻* mice were subjected to medullar aplasia by lethal whole-body irradiation (9.5 grays). The mice were repopulated with an intravenous injection of bone marrow (BM) cells isolated from femurs and tibias of sex-matched C57BL/6 J *Cd36⁻/⁻Card9⁺/⁺* mice or *Cd36⁻/⁻Card9⁻/⁻* littermates. After 4 weeks of recovery, mice were fed a pro-atherogenic high-fat diet containing 15% fat, 1.25% cholesterol, and 0% cholate for 8 weeks.

### Human carotid plaques

Formalin-fixed and paraffin-embedded arterial tissue sections were used after antigen retrieval by heating in microwave oven in Tris/EDTA buffer pH9 (Dako). buffer. For single labeling, CARD9 rabbit polyclonal antibody (Abcam, Cambridge, UK) was used at 1:500 dilution, incubated for 1 h and revealed using ABC-peroxidase technique (Vector Laboratories, Burlingame, CA, USA). For double labeling, the sections were first incubated with mouse monoclonal anti-CD68 antibodies diluted at 1:50 (to detect macrophages) or anti-α smooth muscle actin antibodies diluted at 1:100 (to detect smooth muscle cells), both from Dako-Agilent (Trappes, France). Sections were then incubated with species-specific secondary antibodies (1:500 dilution, 45 min at room temperature) conjugated to AlexaFluor 488 or 594 (Fischer Scientific) and mounted on microscope slides using the Prolong Antifade Diamond kit (Thermofisher). Image acquisition was performed on a laser scanning confocal microscope (Leica TCS SP8, Leica Microsystems).

**Human ethics.** Our research complies with all relevant ethical regulations for all the human samples used. Immunostaining studies were performed on arteries obtained after surgery (Ethical Committee CPP Ile de France 2013-13-19) and written informed consent was obtained.

Protein of plaques were from the Athero-Express study, a longitudinal vascular biobank study in which participants provided written informed consent, and the study was approved by the Medical Ethics Committee of the University Medical Center Utrecht (NL45885.041.13, METC 13/597, Medical Ethical Committee of University). Blood monocytes were obtained from a longitudinal biobank study in which participants provided written informed consent, and the study was approved by the ethical Committee for the Protection of Human Subjects in Biomedical Research (Inserm N° C10-14).

**Pharmacological in vivo treatment.** Eight-week old male *Apoe⁻/⁻Card9⁺/⁺* and *Apoe⁻/⁻Card9⁻/⁻* mice were treated with daily intraperitoneal injections of rapamycin (4 mg/kg body weight) for 6 weeks and were put on a high-fat diet. Eight-week old male *Apoe⁻/⁻Card9⁺/⁺* and *Apoe⁻/⁻Card9⁻/⁻* mice were treated with metformin (300 mg/kg body weight, drinking water) for 6 weeks and were put on a high-fat diet.

### Extent and composition of atherosclerotic lesions

Plasma cholesterol was measured using a commercial cholesterol kit (DiaSys® Cholestérol FS*). Quantification of lesion size was standardized[51]. Briefly, the basal half of the ventricles and the ascending aorta were perfusion-fixed in situ with 4% paraformaldehyde, then transferred to a PBS-30% sucrose solution, embedded in frozen OCT and stored at −80 °C. Serial 10-μm sections of the aortic sinus with valves (80 per mouse) were cut on a cryostat[52]. One section out of 5 was used for plaque size quantification after Oil red O staining. In total, 16 sections spanning over 800 μm of the aortic root were used to determine the mean lesion area for each mouse. After PBS flushing, the aorta from the root to the iliac bifurcation was removed and fixed with 10% neutral-buffered formalin. After thorough PBS washing, the adventitial tissue was removed and the aorta was longitudinally opened to expose the luminal surface for *en-face* visualization of atherosclerotic lesions after Oil Red O staining. Quantification of Oil Red O positive surface area was performed by a blinded operator. Aortic collagen content was detected using Sirius red staining. Necrotic core surface was quantified after Masson's Trichrome staining. At least 4 sections per mouse were examined for each immunostaining, and appropriate negative controls were used. Morphometric studies were performed using Histolab software (Microvisions)[53]. For immunostaining on mouse atherosclerotic plaques, we used antibodies raised against Card9 (AA 274-530), MOMA-2 (macrophage detection, MAB1852, Merck Milllipore®) and CD3 (T cell detection, A0452, Dako®)[53]. TUNEL (Terminal dUTP nick end-labeling) staining was performed using In Situ Cell Death Detection Kit (histochemistry staining) and TMR Red kit (Fluorescent staining) (Roche). The investigators were blinded to group allocation during data collection and analysis.

### Splenocyte culture

Splenocytes were cultured in RPMI 1640 supplemented with Glutamax, 10% fetal calf serum (FCS), 0.02 mM β-mercaptoethanol and antibiotics. For cytokine measurements, splenocytes were stimulated with LPS (1 μg/ml) and IFN-γ (100 UI/ml) for 24 h. IL-1β, IL-10 and TNF-α production in the supernatants were measured using specific ELISA immunoassay kits (BD Biosciences).

### Spleen cell recovery and purification

Spleen cells were purified according to standard protocols as follows. CD4⁺ T cells were negatively selected using a cocktail of anti-CD8a, anti-CD11b, anti-CD45R, anti-DX5, anti-ter 119 antibody-coated magnetic beads, yielding CD4⁺ cells with >95% purity (Miltenyi Biotech). CD11c⁺ cells were positively selected with biotin-conjugated anti-CD11c mAb (7D4, PharMingen) and captured with streptavidin microbeads (Miltenyi Biotec) followed by 2 consecutive magnetic cell separations using LS columns (Miltenyi Biotec), yielding CD11c⁺ cells with >80% purity.

## CD4⁺ T cell culture and cytokine assays

Cells were cultured in RPMI 1640 supplemented with Glutamax, 10% FCS, 0.02 mM β-mercaptoethanol and antibiotics. For cytokine measurements, CD4⁺ T cells were cultured at $1 \times 10^5$ cells/well for 48 h in anti-CD3-coated microplates (5 μg/ml). In some experiments, CD4⁺ T cells were stimulated with purified soluble CD3-specific antibody (1 μg/ml, Pharmingen) in the presence of antigen-presenting cells that were purified on CD11c-coated magnetic beads (Miltenyi Biotech). Secretion of IL-17A, IL-22, IL-10, and IFN-γ secretion in the supernatants was measured using analyte-specific ELISAs (BD Biosciences and R&D Systems). T cell proliferation was measured using the Quick Cell proliferation Assay Kit II (Abcam).

**Macrophage experiments.** Primary macrophages were derived from mouse BM cells (BMDMs). Tibias and femurs of *C57Bl6/J* male mice were dissected and their marrow flushed out. Cells were grown in RPMI 1640 medium, 10% FCS, and 15% Macrophage–Colony-Stimulating Factor (M-CSF)-rich L929-conditioned medium for 7 days at 37 °C. To analyze oxidized LDL uptake, BMDMs were exposed to human oxidized LDL (25 μg/ml) for 24 h (see oxidation method below). Cells were then washed, fixed and stained using Bodipy (493/503, Thermofischer Scientific D3922). Foam cells were quantified blindly on 6–8 fields and the mean was recorded. To analyze apoptosis susceptibility, macrophages were incubated with OxLDL (200 μg/ml) for 6, 12, and 24 h. Apoptosis was determined by independent experiments using Annexin V- (FITC) apoptosis detection kit with 7-AAD (PerCP) (BD Biosciences) according to the manufacturer's instructions. Intracellular cholesterol (total and ester) quantification was done using Amplex® Red Cholesterol Assay Kit (Invitrogen A12216). For cytokine measurements, BM-derived macrophages were exposed to human oxLDL (25 μg/ml) and stimulated with LPS (1 μg/ml) for 24 h. Cytokine production in the supernatants were measured using specific ELISA immunoassay kits (BD Biosciences).

**Immunofluorescence on macrophages.** BM-Derived macrophages were fixed in ice-cold methanol, then washed twice in PBS. Cells were permeabilized with TBS + 0,1% triton for 10 min, then non-specific epitopes were blocked in TBS + 0,1% tween + 3% BSA for 20 min. Primary antibodies diluted in blocking buffer were incubated overnight at 4 °C. The following antibodies were used: Guinea-pig anti-P62 (Progen, GP62-C) and Rabbit anti-LC3B (Cell signaling technology, #43566). After washes in TBS + 0,1% tween, cells were incubated with alexa-568 donkey anti-guinea-pig and alexa-488 donkey anti-rabbit secondary antibodies (Invitrogen) for 2 h at room temperature. After incubation, cells were washed, nuclei were stained with Hoechst and slides were mounted with ibidi fluorescent mounting medium. Images were acquired on a Leica SP8 confocal with lightning super resolution module.

Quantifications were done with ImageJ (NIH) using semi-automatic macros. Briefly, cells were manually circled to create ROI, then for each staining a threshold was applied and quantification of the number of particles, mean area of particles and area of ROI was measured using "Analyze particle". For P62, inclusion bodies were defined as particles with area >100px. Macros are available upon request. Experiments were done in 4 replicates. For LC3+ dots per cell, 25 to 47 cells per condition were quantified and for P62+ inclusion bodies, 20 to 53 cells per condition were quantified.

**Cholesterol efflux assays.** BMDMs were obtained by differentiation of BM cells in Dulbecco's modified Eagle medium (DMEM) supplemented with 10% fetal bovine serum, 2 mM glutamine, 20% L929 cell-conditioned media (as a source of M-CSF), and penicillin-streptomycin for 5 days. BMDMs were loaded with 50 μg/ml [3H]cholesterol-labeled acetylated LDL (acLDL, 1 μCi/mL) for 48 h in serum-free DMEM supplemented with 50 mM glucose, 2 mM glutamine, 0.2% BSA (RGGB),

and 100 μg/ml penicillin / streptomycin. The labeling medium was then removed and cells were washed twice in PBS and then equilibrated in RGGB for additional 16–24 h. To measure cholesterol efflux, cells were incubated 4 h at 37 °C in the presence of 60 μg/ml lipid-free apoAI (Sigma), or 30 μg/ml HDL-PL (density = 1.063–1.21 g/ml), isolated from normolipidemic plasma by preparative ultracentrifugation, as cellular cholesterol acceptor. Finally, culture media was harvested and cleared of cellular debris by brief centrifugation. Fractional cholesterol efflux (expressed as a percentage) was calculated as the amount of radio-label detected in the supernatants divided by total radio-label in each well (radioactivity in the supernatant plus radioactivity in the cells) obtained after lipid extraction from cells in a mixture of 3:2 hexane:isopropanol (3:2 vol/vol). The background cholesterol efflux obtained in the absence of any acceptor was subtracted from the efflux obtained with samples.

**Flow cytometry.** Blood and spleen samples were collected at sacrifice for analysis of leukocyte subsets. Myeloid cells were identified as CD45+ CD11b+. Monocytes were identified as CD11b+ CD115+. Among them, classical monocytes were Gr1^high (or Ly6C^high) and non-classical monocytes were Gr1^low (or Ly6C^low). Neutrophils were identified as CD11b+ CD115-Gr1+ (or CD11b+ CD115-Ly6G+). B220 + IgM+ B lymphocytes, CD4+ and CD8+ T lymphocyte subsets were also analyzed. Antibodies raised against CD11b, CD115, Gr1 (Ly6C and G), B220, CD4, CD8a, NK1.1, CD45, F4/80, CD3ε, MHC II, IgM, CD11c, and CD36 were used for immunostaining and are listed in supplementary data 1.

Forward scatter (FSC) and side scatter (SSC) were used to gate live cells excluding red blood cells, debris, and cell aggregates in total blood cells and splenocytes preparations. Cells were acquired using a BD LSRII Fortessa flow cytometer (BD Biosciences) and analyzed with FlowJo™ (TreeStar, Inc.).

**Quantitative real-time PCR.** RNA extraction was done either with Trizol or with Qiagen columns (RNeasy MiniSpin Columns) using a polytron (T25 basic, IKA, Labortechnik). The phase containing RNAs was then recuperated and washed with molecular biology water. RNA quality control and concentration were performed using Nanodrop 2000 (Thermofisher scientific). Reverse transcription was done following manufacturer instruction [kit QuantiTect Reverse Transcription (Qiagen)]. Real-time fluorescence monitoring was performed with the Applied Biosystems, Step One Plus Real-Time PCR System with Power SYBR Green PCR Master Mix (Eurogentec). qPCR was performed in triplicate for each sample. GAPDH cycle threshold was used to normalize gene expression: (F: 50 -CGTCCCGTAGACAAAATGGTGAA-30; R: 50 - GCC GTGAGTGGAGTCATACTGGAACA-30). Relative expression was calculated using the 2-delta-delta CT method followed by geometric average, as recommended.

The following primer sequences were used: *Card9* (F: 5′- GAC CCT CTT AGT CCC AAT CTG -3′; R: 5′- CTC GTC GTC ATT CTC ATA GTC TG -3′), *Mrs1* (F: 5′- CCG TGA ATC TAC AGC AAA GCA -3′; R: 5′- CCC AGT CCT TCA GTC TGA GG -3′), *Scarb1* (F: 5′- CCT CCT GTT GCT GGT GCC CAT CAT -3′; R: 5′- GCA CTG GTG GGC TGT CCG CTG AGA -3′), *Cd36* (F: 5′- TGG CCA AGC TAT TGC GAC ATG ATT A -3′; R: 5′- CGG GGA TTC CTT TAA GGT CGA TTT C -3′), *Abcg1* (F: 5′- GAC AGC CAT CCC CGT CCT GCT CTT -3′; R: 5′- CTC CCG CAG GAT GGC CTC TGA CTT -3′), and *Abca1* (F: 5′- GGC GGA CCT CCT GTG GTG TTT -3′; R: 5′- GAA TCT CCG GGC TTT AGG GTC CAT -3′).

**Transcriptomic analysis on human monocytes.** RNA sequencing libraries were prepared from 100 to 200 ng of total RNA using the Illumina® Stranded Total RNA Prep, Ligation with Ribo-Zero Plus library preparation kit, which allows performing a strand specific sequencing. This protocol includes a first step of enzymatic depletion of abundant transcripts from multiple species (including human cytoplasmic & mitochondria rRNA, mouse rRNA, rat rRNA, bacteria

Gram +/- rRNA, human beta globin transcripts) using specific probes. cDNA synthesis was then performed and resulting fragments were used for dA-tailing followed by ligation of RNA Index Anchors. PCR amplification with indexed primers (IDT for Illumina RNA UD Indexes) was finally achieved, with 13 cycles, to generate the final cDNA libraries. Individual library quantification and quality assessment were performed using Qubit fluorometric assay (Invitrogen) with dsDNA HS (High Sensitivity) Assay Kit and LabChip GX Touch using a High Sensitivity DNA chip (Perkin Elmer). Libraries were then equimolarly pooled and quantified by qPCR using the KAPA library quantification kit (Roche). Sequencing was carried out on the NovaSeq 6000 instrument from Illumina using paired-end 2 × 100 bp, to obtain around 100 million clusters (200 million raw paired-end reads) per sample. Raw and normalized counts are provided in Supplementary data.

We performed the gene set enrichment analysis using cluster-Profiler v4.0.5[54] with selected pathways from Gene Ontology (GO) and Kyoto Encyclopedia of Genes and Genomes (KEGG) databases, and Benjamini-Hochberg correction was applied. Used keywords were: "apopto*", "atheroscleros*", "NF-kappa B", "TNF". All pathways with adjusted p-value below 0.05 were considered as significantly enriched. For each category, median expression of all gene included in core enrichment were calculated for each patient before plotting the heatmap.

**Single-cell analysis of Card9/CARD9 expression patterns.** Single-cell RNA sequencing (scRNA-seq) datasets of immune cells from mouse aortic atherosclerotic plaques, reported in various publications[55–61], (see ref. 34 for details), were pooled and integrated using canonical correlation analysis (CCA) in Seurat v4.3.0[62,63]. All datasets used for analysis were pre-processed as follows: cells containing >200 detected genes, and genes detected in at least 3 cells were included in the analysis using the 'CreateSeuratObject' function with 'min.features = 200' and 'min.cells=3'. Quality control filtering was further performed to remove dead/damaged cells with a high proportion of mitochondrial transcripts, and outlier cells with high UMI numbers (probable doublets/mutliplets). For mitochondrial transcripts, a < 5% cutoff was applied for all datasets, except for the data from Williams et al.[60] and from Gil-Pulido et al.[61], where <7.5% and <10% cutoffs were applied, respectively. Pre-processing code for aortic leukocyte datasets can be found as supplemental files of Zernecke et al.[34]. All data were log normalized using the 'NormalizeData' function in Seurat with default parameters. Data integration was performed using a canonical CCA workflow in Seurat with default parameters. After CCA integration, data were scaled using 'ScaleData', and principal component analysis was performed using 'RunPCA'. Dimensional reduction was performed using 'RunUMAP' with 30 principal components. Clustering was performed using 'FindNeighbors' with 30 principal components, and 'FindClusters' with a resolution of 0.4. Immune cell annotations are based on Zernecke's works[34,64].

Mouse scRNA-seq data given in Supplementary Fig. 2 were obtained from Pan et al.[18] and downloaded from Gene Expression Omnibus GSE155513, pre-processed in cellranger-6.1.2, and further analyzed in Seurat v4.3.0[63]. We used data from *Ldlr*[-/-] mice fed normal chow or a western diet for 8, 16 or 26 weeks (i.e. the following data from Gene Expression Omnibus GSE155513: GSM4705592, GSM4705593, GSM4705594, GSM4705595, GSM4705596, GSM4705597, GSM4705598, GSM4705599). Individual datasets were pre-processed with quality control filtering in Seurat: cells containing >200 detected genes, and genes detected in at least 3 cells were included in the analysis using the 'CreateSeuratObject' function with 'min.features = 200' and 'min.cells = 3'. Quality control filtering was further performed to remove dead/damaged cells with a high proportion of mitochondrial transcripts (>10%), and outlier cells with high UMI numbers. All data were log normalized using the 'NormalizeData'

function in Seurat with default parameters. Data were pooled and batch corrected using Harmony[65] within Seurat. 2000 highly variable genes were identified using 'FindVariableFeatures' (with selection.method = "vst"). Data were scaled using 'ScaleData' with default parameters, and principal component analysis performed using 'RunPCA' with default parameters, and batch corrected using 'RunHarmony' with default parameters. Dimensional reduction was performed using 'RunUMAP(reduction = "harmony", dims = 1:20)', and clustering was performed at a 0.4 resolution using 'FindNeighbors(reduction = " harmony", dims = 1:20)' followed by 'FindClusters(resolution = 0.2)'. Positive marker genes for each cluster were identified using 'FindAllMarkers'.

Human scRNA-seq data given in Fig. 7 were obtained from total cells of human atherosclerotic coronary arteries[33] and analyzed in Seurat v3[62] starting from the author provided cell-count matrix (downloaded from Gene Expression Omnibus GSE131778). Cells containing <200 detected genes were excluded, and genes detected in at least 3 cells were included in the analysis using the 'CreateSeuratObject' functions with 'min.features = 200' and 'min.cells = 3'. Further quality control filtering was performed and cells with >5% mitochondrial transcripts were excluded, as well as cells with outlier number of UMIs (nCount_RNA > 15,000). A total of 10,934 cells were analyzed. As a pre-analysis indicated a substantial patient-driven batch effect, we performed batch correction using Harmony[65] within Seurat, considering each patient as an independent sample. Data were normalized using the 'NormalizeData' function in Seurat with default parameters. In all, 2000 highly variable genes were identified using 'FindVariableFeatures' (with selection.method = "vst"). Data were scaled using 'ScaleData' with default parameters, and principal component analysis performed using 'RunPCA' with default parameters, and batch corrected using 'RunHarmony' with default parameters. Dimensional reduction was performed using 'RunUMAP(reduction = "harmony", dims = 1:20)', and clustering was performed at a 0.4 resolution using 'FindNeighbors(reduction = "harmony", dims = 1:20)' followed by 'FindClusters(resolution = 0.4)'. Positive marker genes for each cluster were identified using 'FindAllMarkers'. Cell type annotation was performed based on expression of known cell lineage markers, and on cluster annotations in Wirka et al.[33]. In Supplementary Fig. 15e–g, the expression of *CARD9* was examined in scRNA-seq data of mononuclear phagocytes from human coronary and carotid arteries in an integrated dataset described in details in Zernecke et al.[34].

**Western blot.** Proteins from BMDMs were extracted by pipetting in ice-cold lysis buffer (NaCl 150 mM, HEPES 20 mM, EDTA 1 mM, EGTA 1 mM, NP-40 0.25% (Vol/Vol)) supplemented with protease (1 tab/10 ml, Roche) and phosphatase ($Na_4VO_3$ 2 mM) inhibitors). Protein extracts were briefly sonicated (5 pulses, amplitude 20, Vibracell 75021) and quantified by BCA (Thermofisher Scientific). Equal protein amounts were separated by SDS-PAGE (10% acrylamide) and transferred onto nitrocellulose membrane (0.2 µm, Biorad). For LC3b detection, SDS-PAGE separated proteins were transferred onto polyvinylidene difluoride membranes.

Membranes were then blocked with Tris Buffered Saline supplemented with 0.1% Tween-20 (TBST) and 2% BSA (2 h at room temperature), then incubated with primary antibodies (overnight, 4 °C) diluted following the manufacturer's recommendations. After three washes in TBST, membranes were incubated with species-specific horseradish peroxidase-conjugated secondary antibodies (1:8000 dilution, 45 min at room temperature). After three washes in TBST, the peroxidase activity was detected using Clarity Western ECL Substrate (Biorad) using Cytiva's ImageQuant Fluor 800. The migration position of transferred proteins was compared to the PageRuler Prestain Protein Ladder (10 to 170 kDa, Thermo Fisher Scientific). Densitometric analysis was performed using ImageJ software (NIH). Phosphorylated protein signals were normalized on total protein levels, whereas non-

phosphorylated proteins were normalized on β-actin protein levels. Primary and HRP-coupled secondary antibodies used for immunoblotting experiments are listed in supplementary data 3. Uncropped Western blots are available as supplementary material.

### Microbiota analysis

**Stool collection and DNA extraction.** Fecal samples were homogenized and 0.2 g aliquots were stored at −80 °C for further analysis. DNA was extracted from fecal samples using a multi-step protocol[66]. Briefly, the feces samples were weighed and then resuspended for 10 min at room temperature in 250 µl of 4 M guanidine thiocyanate in 0.1 M Tris (pH 7.5) (Sigma-Aldrich) and 40 µl of 10% N-lauroyl sarcosine (Sigma-Aldrich). After the addition of 500 µl of 5% N-lauroyl sarcosine in 0.1 M phosphate buffer (pH 8.0), the 2-ml tubes were incubated at 70 °C for 1 h. One volume (750 ml) of a mixture of 0.1- and 0.6-mm-diameter silica beads (Sigma-Aldrich) (sterilized by autoclaving) was added, and the tube was shaken at 6.5 m/s three times for 30 s each in a FastPrep (MP Biomedicals) apparatus. Polyvinylpolypyrrolidone (15 mg) was added to the tube, which was then vortexed and centrifuged for 5 min at $20,000 \times g$. After recovery of the supernatant, the pellets were washed with 500 µl of TENP (50 mM Tris (pH 8), 20 mM EDTA (pH 8), 100 mM NaCl, 1% polyvinylpolypyrrolidone) and centrifuged for 5 min at $20,000 \times g$, and the new supernatant was added to the first supernatant. The washing step was repeated two times. The pooled supernatant (about 2 ml) was briefly centrifuged to remove particles and then split into two 2-ml tubes. Nucleic acids were precipitated by the addition of 1 volume of isopropanol for 10 min at room temperature and centrifugation for 10 min at $20,000 \times g$. Pellets were resuspended and pooled in 450 µl of 100 mM phosphate buffer, pH 8, and 50 ml of 5 M potassium acetate. The tube was placed on ice overnight and centrifuged at $20,000 \times g$ for 30 min. The supernatant was then transferred to a new tube containing 20 µl of RNase (1 mg/ml) and incubated at 37 °C for 30 min. Nucleic acids were precipitated by the addition of 50 µl of 3 M sodium acetate and 1 ml of absolute ethanol. The tube was incubated for 10 min at room temperature, and the nucleic acids were recovered by centrifugation at $20,000 \times g$ for 15 min. The DNA pellet was finally washed with 70% ethanol, dried, and resuspended in 100 µl of Tris−EDTA (TE) buffer.

**Sequencing.** Microbiota analysis was performed by amplicon sequencing of the V3-V4 region of the 16 S ribosomal RNA gene. This region was amplified using the following primers – 16 S sense 5′-TACGGRAGGCAGCAG-3′ and anti-sense 5′-CTACCNGGGTATC-TAAT-3′ – according to an optimized and standardized 16 S amplicon library preparation protocol (Metabiote, GenoScreen, Lille, France). Briefly, PCR of the 16 S DNA was performed with 5 ng of genomic DNA according to the manufacturer's protocol (Metabiote), with bar-coded primers (Metabiote MiSeq Primers) to a final concentration of 0.2 µmol/l, with an annealing temperature of 50 °C for 30 cycles. Purification of the PCR products was performed with Agencourt AMPure XP-PCR purification system (Beckman Coulter, Brea, CA, USA), and quantified following the manufacturer's instructions. The samples were multiplexed at equal concentrations. Sequencing was performed on an Illumina MiSeq platform (Illumina, San Diego, CA, USA) using a 250 bp paired-end sequencing protocol at GenoScreen. Raw paired-end reads were subjected to the following processes: (1) quality filtering using the PRINSEQ-lite PERL script[67], by truncating the bases from the 3′ end, that did not exhibit a quality <30, based on the Phred algorithm and (2) searching for and removing both forward and reverse primer sequences using CutAdapt, with no mismatches allowed in the primer sequences. Only sequences where perfect matching forward and reverse primers were detected were included.

**16S sequence analysis.** Sequences were quality filtered using the dada2 software package (version 1.12.1)[68] in the R programming language (R version 3.6.1) to produce amplicon sequence variants (ASVs). Taxonomic classification was performed using the Silva reference database (version 132)[69]. Bacterial ASVs that could not be assigned to Phylum-level taxonomy were excluded. Alpha diversity was estimated using the number of observed species and the Shannon diversity index. Raw sequence data are accessible in the Sequence Read Archive (accession number pending). Beta diversity analysis was performed on proportion-normalized data using the Bray-Curtis index. Assessment for significant differences between clusters was performed using PERMANOVA with the adonis function in the vegan package (version 2.5-6) in R with 99999 permutations.

Differential abundance was tested using linear discriminant analysis with effect size (Lefse) using default settings[21].

**LDL isolation and oxidation.** LDL from normal human pooled sera was prepared by ultracentrifugation and dialyzed against PBS containing 100 µM EDTA. The LDL pool was then diluted to 2 g/l with PBS into a final volume of 3 ml. LDLs were mildly oxidized by UV-C for 2 h in the presence of 5 µM $CuSO_4$[70]. Oxidized LDL contained 4.2–7.4 nmoles of TBARS (thiobarbituric acid-reactive substances)/µg apoB. Relative electrophoretic mobility (REM) and 2,4,6-trinitrobenzenesulfonic acid (TNBS) reactive amino groups were 1.2–1.3 times and 85–92% of native LDL, respectively.

### Statistical analysis

Graphs and statistical analyses were performed using Prism software (Graphpad). Values are expressed as mean ± s.e.m. Differences between values were examined using the nonparametric two-tailed Mann–Whitney, Kruskal–Wallis tests when appropriate and were considered significant at $P < 0.05$.

### Reporting summary

Further information on research design is available in the Nature Portfolio Reporting Summary linked to this article.

## Data availability

The authors declare that the data supporting the findings of this study are available within the paper and its supplementary information files. All the raw data generated in this study are provided in the Source Data file. RNA-sequencing data of blood monocytes generated for this report has been deposited in Gene Expression Omnibus (https://www.ncbi.nlm.nih.gov/geo/query/acc.cgi?acc=GSE221782). Mouse scRNA-seq data were downloaded from Gene Expression Omnibus GSE155513[18], GSM4705592[18], GSM4705593[18], GSM4705594[18], GSM4705595[18], GSM4705596[18], GSM4705597[18], GSM4705598[18], GSM4705599[18], and GSE131780[33]. Microbiota 16s RNA data are accessible with the following link https://www.ncbi.nlm.nih.gov/sra/PRJNA986053. Human carotid atherosclerosis scRNA-seq data were downloaded from https://figshare.com/s/c00d88b1b25ef0c5c788[71]. Human coronary atherosclerosis scRNA-seq data were downloaded from Gene Expression Omnibus GSE131776[33] and GSE131778[33]. Source data are provided with this paper.

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

## Acknowledgements

This work was supported by Inserm (H.A.O. and Z.M.), Nouvelle Société Française d'Athérosclérose (Y.Z.), the Fondation pour la Recherche Médicale (H.A.O. and S.T.), la Fondation Lefoulon-Delalande (Y.S.Z.), la Fondation de l'avenir, The European Research Council (Z.M.), and the British Heart Foundation (Z.M.). C. Cochain was supported by the Interdisciplinary Center for Clinical Research (IZKF, Interdisziplinäres Zentrum für Klinische Forschung), University Hospital Würzburg (Project IZKF-E-353). C.C., A.E.S., and A.Z. are supported by the Deutsche Forschungsgemeinschaft (DFG, German Research Foundation) Project-ID 453989101-SFB1525. S.C.M. received funding from the European Union's Horizon 2020 research and innovation program under the Marie Sklodowska-Curie grant agreement No 846519 and by the Fondation Lefoulon-Delalande, Paris, France. High-throughput sequencing was performed by the ICGex NGS platform of the Institut Curie supported by the grants ANR-10-EQPX-03 (Equipex) and ANR-10-INBS-09-08 (France Génomique Consortium) from the Agence Nationale de la Recherche ("Investissements d'Avenir" program), by the ITMO-Cancer Aviesan (Plan Cancer III) and by the SiRIC-Curie program (SiRIC Grant INCa-DGOS-465 and INCa-DGOS-Inserm_12554). Data management, quality control, and primary analysis were performed by the Bioinformatics platform of the Institut Curie." A.P. and M.M. were supported by the French National Research Agency (ANR) under the "Investments for the future" program (ANR-10-IAHU-01), the ANR-FNS LTh-MSMD-CMCD (ANR-18-CE93-0008-01), the Integrative Biology of Emerging Infectious Diseases Laboratory of Excellence (ANR-10-LABX-62-IBEID), and the National Institute of Allergy and Infectious Diseases of the NIH (grant no. R01AI127564).

## Author contributions

Study concept and design by Y.Z., M.V., J.R.L., J.J., I.S.Z., A.L., X.Z., W.L., M.G., R.A.R., L.L., P.B., C.G., M.D., Me.M., C.C., A.Z., A.E.S., Mi.M., J.S.S., A.T., Z.M., S.T., O.L., C.V., S.C., H.S., and H.A.O. Acquisitions of data by Y.Z., M.V., J.R.L., J.J., I.S.Z., A.L., X.Z., W.L., M.G., R.A.R., L.L., C.G., M.D., Me.M., Mi.M., C.C., O.L., C.V., S.C., H.S., and H.A.O. Drafting of the manuscript by Y.Z., C.C., O.L., A.T., Z.M., S.T., S.C., H.S., and H.A.O. Critical revision of manuscript by Y.Z., M.V., J.R.L., J.J., I.S.Z., A.L., X.Z., W.L., M.G., R.A.R., L.L., P.B., C.G., M.D., Me.M., A.P., F.L., J.L.C., C.C., A.Z., A.E.S., Mi.M., J.S.S., A.T., Z.M., S.T., O.L., C.V., S.C., H.S., and H.A.O. Statistical analysis by Y.Z., M.D., C.C., and H.A.O.

## Competing interests

The authors declare no competing interests.

## Additional information

¹Université Paris Cité, INSERM U970, Paris Cardiovascular Research Center, Paris, France. ²Sorbonne Université, Paris, France. ³Sorbonne Université, INSERM, Centre de Recherche Saint-Antoine, CRSA, AP-HP, Saint Antoine Hospital, Gastroenterology department, Paris, France. ⁴Inserm UMRS1166, ICAN, Institute of CardioMetabolism and Nutrition, Hôpital Pitié-Salpêtrière (AP-HP), Paris, France. ⁵Department of Anatomopathology, Hôpital Européen Georges Pompidou, AP-HP, Paris, France. ⁶Institut Curie, Cytometry Platform, 75006 Paris, France. ⁷Clinique Saint Gatien Alliance (NCT+), 37540 Saint-Cyr-sur-Loire, France; Institut Necker-Enfants Malades (INEM), Université Paris Cité, INSERM UMR-S1151, CNRS UMR-S8253, 75015 Paris, France. ⁸Laboratory of Human Genetics of Infectious Diseases, Necker Branch, INSERM U1163, Imagine Institute, 75015 Paris, France. ⁹St. Giles Laboratory of Human Genetics of Infectious Diseases, Rockefeller Branch, Rockefeller University, New York, NY, USA. ¹⁰Comprehensive Heart Failure Center Wuerzburg, University Hospital Wuerzburg, Wuerzburg, Germany. ¹¹Institute of Experimental Biomedicine, University Hospital Wuerzburg, Wuerzburg, Germany. ¹²Helmholtz Institute for RNA-based Infection Research (HIRI), Helmholtz-Center for Infection Research (HZI), Wuerzburg, Germany. ¹³Laboratory of Experimental Cardiology, Department of Cardiology, University Medical Center Utrecht, University Utrecht, Utrecht, Netherlands. ¹⁴Division of Cardiovascular Medicine, University of Cambridge, Addenbrooke's Hospital, Cambridge CB2 2QQ, UK. ¹⁵CIC 1436/CARDIOMET, Inserm, 1048 Toulouse, France. ¹⁶University Paris-Saclay, INRAE, AgroParisTech, Micalis Institute, Jouy-en-Josas, France. ¹⁷Paris Center for Microbiome Medicine (PaCeMM) FHU, Paris, France. ¹⁸Medical Intensive Care Unit, Hôpital Saint-Antoine, AP-HP, Sorbonne Université, Paris, France. ¹⁹These authors contributed equally: Marie Vandestienne, Jean-Rémi Lavillegrand. ✉e-mail: hafid.aitoufella@inserm.fr

