## [Peer review file · Nature Communications]

REVIEWER COMMENTS

Reviewer #1 (Remarks to the Author):

The authors report on the protective role of CARD9 in the development of atherosclerosis.

By using various complementary approaches, the authors have identified CARD9 signaling as a key protective pathway. This is partly explained by changes in macrophage CD36-dependent inflammatory responses, lipid uptake and autophagy controlled by CARD9.

The paper reads well and contains an impressive number of experiments and results.

Sometimes I feel there is an overload of data that is either misinterpreted or lacks coherence with other results present in the manuscript. In this case, removing data may help to increase readability of the manuscript.

I do have several questions and suggestions that may help to further improve the manuscript.

1. I am somewhat confused by the results from the two animal models used by the authors. First, the authors state that ex vivo stimulated CD4+ T cells from Apoe^{-/-}Card9^{-/-} show significantly increased production of IFN- γ and IL-17A compared to WT cells. In the second model, Splenic CD4+ T cells isolated from Ldlr^{-/-} Card9^{-/-} mice produced less IL-17A. Could the authors explain these opposite findings? I would also suggest to move results of one of the models to the supplement.

2. Why would CARD9 be upregulated both in early and late plaque development? I guess the authors would explain this as a protective mechanism? What would be the drivers of CARD9 gene expression in macrophages part of the plaque? Their results show that the absence of CARD9 leads to more foam cell formation. However, the human data shows that card9 expression was detected in fatty streak lesions (Figure 8d) and in lipid-rich areas surrounding the necrotic core of advanced atherosclerotic plaques (Figure 8e). How would the authors explain these observations ?

3. The microbiome data seem somewhat out of place. Also the interpretation may require some caution. One can't simply conclude that the microbiome does not play a role based on differences in microbiome composition, yet a similar outcome on atherosclerosis in both models. And why would you want to compare these different animal models ? For example, the second model is using bone marrow transplantation and the irradiation that needs to be performed may have long lasting effects on the microbiome.

4. The following sentence is confusion: We also found increased nuclear translocation of RelB in Card9^{-/-} macrophages (Supplementary Fig.6b), which suggests that the mechanism responsible for CD36 up-regulation observed in the absence of Card9 could be related to ox-LDL signaling deviation towards the non-canonical NF- κ B pathway .

How can you make this assumption only based on two independent observations? It requires more work to establish this link.

5. The authors report the following: Interestingly, we observed that Card9 deficiency led to a phenotypic switch towards a more pro-inflammatory macrophage phenotype characterized by higher secretion of IL-1 β and lower secretion of IL-10 following in vitro stimulation (Figure 5i).

How are the cells stimulated? Was this done with cells loaded with oxLDL or naive macrophages?

6. Why have the authors decided to study autophagy based on the changes in CD36? This is unclear. Also, many of the experiments related to autophagy appear to be performed in WT and KO macrophages not exposed to oxLDL. This hampers interpretation of the results in the context of atherosclerosis.

7. Please be careful once interpreting the results of the in vivo experiment using rapamycin. This is an inhibitor of mTOR, a protein complex with a wide variety of functions well beyond autophagy.

The same holds true when using metformin. This would have various different effects far beyond autophagy alone.

Overall, based on the outcomes of these experiments, one can't conclude on a potential role of autophagy.

-The authors state that 'Formalin-fixed and paraffin-embedded arterial tissue sections were used after antigen retrieval by heating in microwave oven in buffer pH9'. What is the composition of the buffer? Also, various details related to information related to used kits (for example, how was cDNA made used for Quantitative real-time PCR and what kit was used?) and approaches are missing in the methods section.

-It would help to add labels to figures. Sometimes it is difficult to read the figures, for example in figure 5, left bottom graphs.

Reviewer #2 (Remarks to the Author):

Zhang and coworkers demonstrate a protective role for Card9 in macrophages in atherogenesis, which is independent of the adaptive immune system and mediated via macrophage CD36 and the regulation of autophagy. This is a novel and highly important pathomechanism influencing the development of atherosclerotic lesions. The protective effect of Card9 was suggested to depend on CD36-regulated autophagy. Single-cell analyses on aortas of Apoe^{-/-} mice demonstrate Card9 expression in macrophage populations, excluding other vascular cell types. To functionally address the role of Card9, the authors generated a Apoe^{-/-}Card9^{-/-} mouse line, which did not show altered plasma cholesterol levels but increased aortic sinus lesion size with elevated macrophage accumulation and collagen content. The

atherosclerosis phenotype is mechanistically explained by the upregulation of the scavenger receptor CD36 in Card9^{-/-} macrophages, which is convincingly supported by the presented data and appears plausible. The revealed link of Card9^{-/-} with impaired autophagy in macrophages is another relevant finding that is consistent with the observed atherosclerotic plaque phenotype. The analysis of bone marrow transplants into Cd36^{-/-}Card9^{-/-} mice further strengthens the authors conclusion of macrophage CD36 being the key factor determining the atherosclerotic lesion phenotype of Apoe^{-/-}Card9^{-/-} mice. In my view, the presented clinical data are not required for this mechanistic study. Although the study includes microbiome sequencing analyses, the conclusions made on these data are overstated. To demonstrate causality would require additional experimentation, including microbiota transplantation into germ-free atherosclerosis mouse models. In essence, the presented data provide strong evidence for the suggested pathomechanism and the study represents a significant advance in our current understanding of cardiovascular disease. If thoroughly revised, this study will define macrophage Card9 as a new target that might turn out therapeutically useful in the prevention of atherosclerotic lesion progression.

1. In the introduction (line 104), the reasoning why Card9 was studied should be explained in more detail. Why did the authors put focus on Card9? Furthermore, the introduction lacks a description of the gut microbiota as a relevant modifier of atherosclerosis. There is recent work on germ-free Ldlr^{-/-} and germ-free Apoe^{-/-} mouse models that should be described in the introduction (PMIDs: 29903735; 30397344; 31641089; 32579470), especially since the authors conclude that the microbiota is not involved in the atherosclerotic phenotype of Apoe^{-/-}Card9^{-/-} mice and BM-transplanted Ldlr^{-/-}Card9^{-/-} mice (Fig.3). To become more intelligible, the introduction also lacks a detailed description of the mechanism by which macrophage CD36 transports oxLDL. These points should be addressed by re-writing.

2. It is convincing to demonstrate the role of Card9 in an additional atherosclerosis mouse model. However, to analyze the role of the hematopoietic cell compartment, why was bone marrow transplantation performed in Ldlr^{-/-}Card9^{-/-} mice but not in Apoe^{-/-}Card9^{-/-} mice? This should be explained at the beginning the respective paragraph (line 424).

3. The heading to the results description of Figure 3 and the conclusion on these 16S rRNA gene sequencing data is clearly overstated. The authors did not perform microbiome transplantation of Apoe^{-/-}Card9^{-/-} and Apoe^{-/-}Card9^{+/+} mice into germ-free atherosclerotic mice to exclude a role of the gut microbiota. Therefore, they cannot conclude that the microbiota is not involved. This paragraph requires rewriting.

4. Can the authors explain the increased plasma cholesterol levels in the Apoe^{-/-}Rag2^{-/-}Card9^{-/-} mice, which is inconsistent with the previous data in Apoe^{-/-}Rag2^{-/-} mice? It is important to exclude adaptive immunity but the Apoe^{-/-}Card9^{-/-} plaque phenotype was independent of plasma cholesterol levels.

5. Western blot and/or FACS analyses on CD36 expression, preferentially in plaque resident MOMA-positive macrophage populations, would further strengthen the conclusion of CD36 being involved in the pathomechanism. Immunofluorescence analysis is not ideal for quantification of CD36.

6. In the reviewer's view, the analysis of CARD9-related pathways in human (Figure 8) is not well connected to the very stringent mouse atherosclerosis data. These patient data do not significantly add and they are not strengthening the mechanistic conclusions of this work. Therefore, the analyses on CARD9-deficient patients carrying homozygous c.865C>T mutations should rather be dealt with in a subsequent clinical study.

7. The data presentation of the figures needs to be improved. Furthermore, the discussion part should be written more concisely.

8. Please correct the typo in line 654 (Ldl/-).

9. A detailed scheme on the revealed CARD9-dependent pathomechanism is missing, so the novelty of the concept becomes clear.

Reviewer #3 (Remarks to the Author):

In this paper by Zhang et al. the authors perform both in vivo and in vitro modeling to investigate the role of CARD9 in atherosclerosis. They demonstrate that Card9 deficiency increases atherosclerosis, likely through elevated pro-inflammatory cytokines, lipid uptake, increased cell death and defective autophagy, which was reversed by metformin and rapamycin. Card9 deficiency in macrophages upregulated Cd36 and effects on autophagy could be reversed in absence of Cd36. Finally, they perform bulk and single-cell RNA-seq analyses to confirm the pathogenic signature induced with CARD9 deficiency. The comments below are mostly focused on the transcriptomic studies and are meant to improve the quality of the study.

Major comments

1. The authors performed an integration of murine atherosclerosis single-cell datasets; however, they do not mention any of the QC parameters (e.g., number of unique genes, UMIs, % read mapped to mitochondrial genome) used for processing these datasets. It should be noted whether the same QC parameters from each separate study were used or if there were additional metrics used. Also, there is no information about how the single-cell data was normalized. This is critical as these steps need to be standardized before integration.
2. Similarly, the methods for analyzing the human coronary artery scRNA-seq data also lacks sufficient details regarding QC, normalization, dimensionality reduction, clustering, and cell type annotation steps.
3. In Fig 1a, there is a population of SMCs that seems to be at the intersection of the main SMC and T cell cluster. Since these distinct cell types would be expected to be further from each other in the UMAP space it is possible that this population of SMCs could represent multiplets from the previous publications. Similarly, there is not a single distinct cluster of B cells and the cells closer to myeloid cells might reflect doublets. The authors should carefully inspect the individual datasets before and after integration to reduce this technical artifact.
4. Is CARD9 specifically expressed in anti or pro-inflammatory macrophages? This would be important to discern as the authors make the claim that Card9 is protective and use the broad term of macrophages.
5. Given that the authors are highlighting the macrophage specific role of CARD9, it would be helpful to show specific macrophage markers that are expressed within the large cluster to support the cell type annotations.
6. In Fig 8, it is confusing that there are two separate clusters labeled as VSMCs that are away from each other. The authors should further inspect these to determine whether these represent SMCs and pericytes separately.
7. Similar to Fig 1, it would be helpful to show markers of human macrophages that are expressed within the large cluster to support the cell type annotations. This could be done with UMAP feature plots or included within Supplementary tables.
8. Regarding the immunostaining in Fig 8, this appears incomplete and there are no general histology stains to assess the lesion stage or location of the staining in D and E. Some quantitation of CARD9 mRNA or protein levels in human athero versus normal arteries would help support the staining results.

Minor comments

1. Some of the text labels in Fig3, Fig 6 and Fig7 are quite small and almost not legible. The text should be increased to be consistent with the other figures.
2. Magnification or ruler should be added to the immunostaining images in Fig 8.

Responses to reviewers

Reviewer #1 (Remarks to the Author):

The authors report on the protective role of CARD9 in the development of atherosclerosis. By using various complementary approaches, the authors have identified CARD9 signaling as a key protective pathway. This is partly explained by changes in macrophage CD36-dependent inflammatory responses, lipid uptake and autophagy controlled by CARD9. The paper reads well and contains an impressive number of experiments and results. Sometimes I feel there is an overload of data that is either misinterpreted or lacks coherence with other results present in the manuscript. In this case, removing data may help to increase readability of the manuscript.

Response: We would like to thank the reviewer for his/her encouraging comments which helped improve our manuscript.

I do have several questions and suggestions that may help to further improve the manuscript. 1. I am somewhat confused by the results from the two animal models used by the authors. First, the authors state that ex vivo stimulated CD4⁺ T cells from Apoe^{-/-}Card9^{-/-} show significantly increased production of IFN- γ and IL-17A compared to WT cells. In the second model, Splenic CD4⁺ T cells isolated from Ldlr^{-/-} Card9^{-/-} mice produced less IL-17A. Could the authors explain these opposite findings? I would also suggest to move results of one of the models to the supplement.

Response: Indeed, we found different impact of Card9 deficiency on CD4⁺ T cell profiles between Apoe^{-/-} and chimeric Ldlr^{-/-} models. These discrepancies in cytokine production could be due to gender (male Apoe^{-/-} vs female chimeric Ldlr^{-/-}) or background (Apoe^{-/-} vs Ldlr-deficiency) differences. In this regard, several works have reported that apolipoprotein E, per se, has several functions not related to lipid metabolism, which may impact on T cell-mediated immune responses. For example, Apoe is involved in efferocytosis (Cash et al JBC 2012), a key regulator of T cell function (Ait-Oufella Circulation 2007). In addition, it has been reported that Apoe plays a major role in DC function and antigen presentation to T cells (Bonacina et al, Nat Commun 2018). These information have been detailed in the Discussion section (Page 27). We did not investigate in depth how Card9 could modulate T cell immunity, because we found similar findings (acceleration of atherosclerosis in the absence of Card9) in immunocompetent Apoe^{-/-} Card9^{-/-} or immunodeficient Apoe^{-/-}Rag2^{-/-} Card9^{-/-} mice, ruling out an important role of T or B cell immunity in the Card9^{-/-} vascular phenotype. As suggested by the reviewer, graphs depicting cytokine production by CD4⁺ T cells in Apoe^{-/-} and chimeric Ldlr^{-/-} models has been moved to supplemental figures 4F & 8C.

2. Why would CARD9 be upregulated both in early and late plaque development? I guess the authors would explain this as a protective mechanism? What would be the drivers of CARD9 gene expression in macrophages part of the plaque? Their results show that the absence of CARD9 leads to more foam cell formation. However, the human data shows that Card9 expression was detected in fatty streak lesions (Figure 8d) and in lipid-rich areas surrounding the necrotic core of advanced atherosclerotic plaques (Figure 8e). How would the authors explain these observations?

Response: Card9 expression is induced by engagement of fungi receptors (Dectin and Mincle), as well as by Tlr4 activation (Gross et al. Blood 2008). To address the reviewer's concern, we performed in vitro additional experiments in BM-derived macrophages. We confirmed that oxLDL, which binds to Tlr4, induced Card9 expression in macrophages, but to a lesser extent than Curdlan, a component of fungi membrane. This result has been added as supplementary fig 3.

Overall, using several models we showed that Card9 plays a protective role in atherosclerosis development. Card9 expression in atherosclerotic plaques at both early and late stages is not in contradiction with murine models with Card9 deficiency. Indeed, we believe that Card9 expression in

fatty streak lesions and in lipid-rich areas surrounding the necrotic core of advanced atherosclerotic plaques *is likely a regulatory compensatory mechanism to limit foam cell formation.*

3. The microbiome data seem somewhat out of place. Also, the interpretation may require some caution. One can't simply conclude that the microbiome does not play a role based on differences in microbiome composition, yet a similar outcome on atherosclerosis in both models. And why would you want to compare these different animal models? For example, the second model is using bone marrow transplantation and the irradiation that needs to be performed may have long lasting effects on the microbiome.

Response: We would like to thank the reviewer for this comment. We analyzed gut microbiota in both murine models because: 1. Card9 is well known to be involved in microbiota homeostasis (Sokol et al. Gastroenterology 2013) and 2. gut dysbiosis has been reported to modulate atherosclerosis (Ahmad et al. Am J Physiol Heart Circ Physiol. 2019). To address the reviewer's comment, microbiome data have been moved to supplementary figure 9. Given that we did not perform microbiota transfer experiments, our conclusion about the role of Card9 deficiency-related dysbiosis in our phenotype has been tempered and a mention of this limitation has been added in the Discussion section (Page 30).

4. The following sentence is confusion: We also found increased nuclear translocation of RelB in Card9^{-/-} macrophages (Supplementary Fig.6b), which suggests that the mechanism responsible for CD36 up-regulation observed in the absence of Card9 could be related to ox-LDL signaling deviation towards the non-canonical NF- κ B pathway. How can you make this assumption only based on two independent observations? It requires more work to establish this link.

Response: Thank you for this comment. We agree with the reviewer that this is too preliminary. Therefore, we removed the data on RelB from the revised manuscript.

5. The authors report the following: Interestingly, we observed that Card9 deficiency led to a phenotypic switch towards a more pro-inflammatory macrophage phenotype characterized by higher secretion of IL-1 β and lower secretion of IL-10 following in vitro stimulation (Figure 5i). How are the cells stimulated? Was this done with cells loaded with oxLDL or naive macrophages?

Response: For cytokine measurements, BM-derived macrophages were exposed to oxidized LDL (25 ug/ml) and stimulated with LPS (1 ug/ml) for 24 hours. Cytokine production in the supernatants was measured using specific ELISA immunoassay kits (BD Biosciences). This point has been clarified in the Methods section and detailed in figures' legends.

6. Why have the authors decided to study autophagy based on the changes in CD36? This is unclear. Also, many of the experiments related to autophagy appear to be performed in WT and KO macrophages not exposed to oxLDL. This hampers interpretation of the results in the context of atherosclerosis.

Response: We would like to thank the reviewer for this comment. We decided to investigate the impact of Card9 deficiency on autophagy in the context of atherosclerosis for several reasons. 1. We observed an increase in death susceptibility of Card9-deficient macrophages, which could be due to defective autophagy. 2. We observed an acceleration of atherosclerosis in Card9-deficient animals, which has been reported in murine models with autophagy alteration. 3. Previous works have reported that Card9 interacts with Rubicon, a key regulator of autophagy flux. 4. CD36, which is up related in Card9-deficient macrophages, has been shown to modulate early steps of autophagy induction through the blockade of AMPK phosphorylation (LI et al. J Lipid Res 2019). This point has been clarified in the manuscript (Page 22-23).

All the in vitro experiments (except new figure 5 B) have been done in macrophages exposed to oxLDL. Details have been clarified in the Methods section and in the figures' legends.

7. Please be careful once interpreting the results of the in vivo experiment using rapamycin. This is an inhibitor of mTOR, a protein complex with a wide variety of functions well beyond autophagy. The same holds true when using metformin. This would have various different effects far beyond autophagy alone. Overall, based on the outcomes of these experiments, one can't conclude on a potential role of autophagy.

Response: To address the reviewer's concern, we perform an additional series of in vitro experiments with immunofluorescence staining and confocal analysis. We confirmed impaired autophagy in Card9-deficient macrophages with the accumulation of very large p62+ inclusion bodies (New figure 5B, 5D). In addition, we found that rapamycin and metformin restored in vitro autophagy flux with a strong decrease and almost abolition of these inclusion bodies after treatment (New figure 5D & Figure 6A-B). In addition, rapamycin and metformin abolished both the accumulation of p62 within the lesions (see figure 5 & 6) and the pro-atherogenic effects of Card9 deficiency. However, as we cannot rule out some off target effects of these two drugs, we added a limitation note in the Discussion section (P29).

-The authors state that 'Formalin-fixed and paraffin-embedded arterial tissue sections were used after antigen retrieval by heating in microwave oven in buffer pH9'. What is the composition of the buffer?

Response: We used Tris/EDTA buffer (Dako). This point has been clarified in the Methods section

Also, various details related to information related to used kits (for example, how was cDNA made used for Quantitative real-time PCR and what kit was used?) and approaches are missing in the methods section.

Response: We do apologize about that. The methods have been clarified in the manuscript.

-It would help to add labels to figures. Sometimes it is difficult to read the figures, for example in figure 5, left bottom graphs

Response: The modifications have been made accordingly in previous Figure 5 (Figure 4 in the revised R1 version)

Reviewer #2 (Remarks to the Author):

Zhang and coworkers demonstrate a protective role for Card9 in macrophages in atherogenesis, which is independent of the adaptive immune system and mediated via macrophage CD36 and the regulation of autophagy. This is a novel and highly important pathomechanism influencing the development of atherosclerotic lesions. The protective effect of Card9 was suggested to depend on CD36-regulated autophagy. Single-cell analyses on aortas of Apoe^{-/-} mice demonstrate Card9 expression in macrophage populations, excluding other vascular cell types. To functionally address the role of Card9, the authors generated a Apoe^{-/-}Card9^{-/-} mouse line, which did not show altered plasma cholesterol levels but increased aortic sinus lesion size with elevated macrophage accumulation and collagen content. The atherosclerosis phenotype is mechanistically explained by the upregulation of the scavenger receptor CD36 in Card9^{-/-} macrophages, which is convincingly supported by the presented data and appears plausible. The revealed link of Card9^{-/-} with impaired autophagy in macrophages is another relevant finding that is consistent with the observed atherosclerotic plaque phenotype. The analysis of bone marrow transplants into Cd36^{-/-}Card9^{-/-} mice further strengthens the authors conclusion of macrophage CD36 being the key factor determining the atherosclerotic lesion phenotype of Apoe^{-/-}Card9^{-/-} mice. In my view, the presented clinical data are not required for this mechanistic study. Although the study includes microbiome sequencing analyses, the conclusions made on these data are overstated. To demonstrate causality would require additional experimentation, including microbiota transplantation into germ-free atherosclerosis mouse models. In essence, the presented data provide strong evidence for the suggested pathomechanism and the study represents a significant advance in our current understanding of cardiovascular disease. If thoroughly revised, this study will define macrophage Card9

as a new target that might turn out therapeutically useful in the prevention of atherosclerotic lesion progression.

Response: We would like to thank the reviewer for his/her encouraging comments which helped improve the manuscript.

1. In the introduction (line 104), the reasoning why Card9 was studied should be explained in more detail. Why did the authors put focus on Card9? Furthermore, the introduction lacks a description of the gut microbiota as a relevant modifier of atherosclerosis. There is recent work on germ-free Ldlr^{-/-} and germ-free Apoe^{-/-} mouse models that should be described in the introduction (PMIDs: 29903735; 30397344; 31641089; 32579470), especially since the authors conclude that the microbiota is not involved in the atherosclerotic phenotype of Apoe^{-/-}Card9^{-/-} mice and BM-transplanted Ldlr^{-/-}Card9^{-/-} mice (Fig.3). To become more intelligible, the introduction also lacks a detailed description of the mechanism by which macrophage CD36 transports oxLDL. These points should be addressed by re-writing.

Response: We would like to thank the reviewer for hi/hers comments. The introduction has been modified accordingly and references have been updated.

2. It is convincing to demonstrate the role of Card9 in an additional atherosclerosis mouse model. However, to analyze the role of the hematopoietic cell compartment, why was bone marrow transplantation performed in Ldlr^{-/-}Card9^{-/-} mice but not in Apoe^{-/-}Card9^{-/-} mice? This should be explained at the beginning the respective paragraph (line 424).

Response: After our first observation showing an acceleration of atherosclerosis in Apoe^{-/-} background, we aimed to confirm this in Ldlr^{-/-} mice, which is a another classical murine model of atherosclerosis. We performed BM transplantation in irradiated Ldlr^{-/-} to specifically investigate the hematopoietic compartment. Our results are consistent in both models. This point has been clarified in the manuscript (Page 19)

3. The heading to the results description of Figure 3 and the conclusion on these 16S rRNA gene sequencing data is clearly overstated. The authors did not perform microbiome transplantation of Apoe^{-/-}Card9^{-/-} and Apoe^{-/-}Card9^{+/+} mice into germ-free atherosclerotic mice to exclude a role of the gut microbiota. Therefore, they cannot conclude that the microbiota is not involved. This paragraph requires rewriting.

Response: We agree with the reviewer for his/her comments. Given that we did not perform microbiota transfer experiments, our conclusion about the role of Card9 deficiency-related dysbiosis in the observed phenotype has been tempered. Microbiome data have been moved as supplementary figure 9 and a mention of limitation has been added in the Discussion section (Page 30).

4. Can the authors explain the increased plasma cholesterol levels in the Apoe^{-/-}Rag2^{-/-}Card9^{-/-} mice, which is inconsistent with the previous data in Apoe^{-/-}Rag2^{-/-} mice? It is important to exclude adaptive immunity but the Apoe^{-/-}Card9^{-/-} plaque phenotype was independent of plasma cholesterol levels.

Response: We would like to thank the reviewer for his/her comment. Indeed, we found higher cholesterol plasma levels in Apoe^{-/-}Rag2^{-/-}Card9^{-/-}, compared to Apoe^{-/-}Rag2^{-/-}Card9^{+/+} mice. However, the increase was very modest (+17%) when compared to the acceleration of atherosclerosis observed in the aortic sinus (+50%) and along the thoraco-abdominal aorta (+69%). In addition, given that Card9 deficiency led to an acceleration of atherosclerosis in both Apoe^{-/-} and Ldlr^{-/-} background without any differences in plasma cholesterol levels, we are reassured that the pro-atherogenic effects of Card9 deficiency were not due by changes in plasma cholesterol levels. This point has been clarified in the Results section.

5. Western blot and/or FACS analyses on CD36 expression, preferentially in plaque resident MOMA-positive macrophage populations, would further strengthen the conclusion of CD36 being involved in the pathomechanism. Immunofluorescence analysis is not ideal for quantification of CD36.

Response: To address the reviewer's concern, we performed an additional series of experiments. Apoe^{-/-}Card9^{+/+} and Apoe^{-/-}Card9^{-/-} mice were fed a high fat diet during 6 weeks and macrophages were analyzed by flow cytometry after enzymatic digestion. We found higher numbers of CD45⁺Ly6GCD11b⁺CD64⁺ macrophages expressing CD36 in the vascular wall of Card9-deficient mice, in agreement with our in vitro and results and the qPCR analysis. These results have been added as a new supplemental figure 10B.

6. In the reviewer's view, the analysis of CARD9-related pathways in human (Figure 8) is not well connected to the very stringent mouse atherosclerosis data. These patient data do not significantly add and they are not strengthening the mechanistic conclusions of this work. Therefore, the analyses on CARD9-deficient patients carrying homozygous c.865C>T mutations should rather be dealt with in a subsequent clinical study.

Response: We would like to thank the reviewer for his/her comment. Indeed, human data were obtained from monocytes, whereas our experimental studies were performed in macrophages. However, the inflammatory signatures in CARD9-deficient monocytes are consistent with the cytokine profiles obtained in murine models (i.e.; Il1b overproduction), as well as impaired pathways identified in Apoe^{-/-} and Ldlr^{-/-} mice, including apoptosis, autophagy and atherosclerosis.

7. The data presentation of the figures needs to be improved. Furthermore, the discussion part should be written more concisely.

Response: We would like to thank the reviewer for his/her in-depth analysis of our work. Based on his/her comments, data about cytokine production by CD4⁺ T cells in Apoe^{-/-} (Fig.1) and Ldlr^{-/-} (Fig.2) mice have been moved to supplemental figures. Also, microbiome data have been moved to supplementary figure 9. Finally, the discussion has been modified with a mention of additional limitations, as required by the reviewers.

8. Please correct the typo in line 654 (Ldl^{-/-}).

Response: The modification has been done accordingly

9. A detailed scheme on the revealed CARD9-dependent pathomechanism is missing, so the novelty of the concept becomes clear.

Response: As suggested, a summary cartoon has been added (Supplementary Figure 16)

Reviewer #3 (Remarks to the Author):

In this paper by Zhang et al. the authors perform both in vivo and in vitro modeling to investigate the role of CARD9 in atherosclerosis. They demonstrate that Card9 deficiency increases atherosclerosis, likely through elevated pro-inflammatory cytokines, lipid uptake, increased cell death and defective autophagy, which was reversed by metformin and rapamycin. Card9 deficiency in macrophages upregulated Cd36 and effects on autophagy could be reversed in absence of Cd36. Finally, they perform bulk and single-cell RNA-seq analyses to confirm the pathogenic signature induced with CARD9 deficiency. The comments below are mostly focused on the transcriptomic studies and are meant to improve the quality of the study.

Response: We would like to thank the reviewer for his/her comments that helped improve the manuscript.

Major comments

1. The authors performed an integration of murine atherosclerosis single-cell datasets; however, they do not mention any of the QC parameters (e.g., number of unique genes, UMIs, % read mapped to mitochondrial genome) used for processing these datasets. It should be noted whether the same QC parameters from each separate study were used or if there were additional metrics used. Also, there is no information about how the single-cell data was normalized. This is critical as these steps need to be standardized before integration.

*In **Figure 1A**, single-cell RNA-seq datasets of aortic immune cells from various publications (see Zernecke, Cardiovascular Research 2022, PMID: 36190844, for details) and from total aortic cells in atherosclerotic mice (Gil-Pulido, Cardiovascular Research 2022, PMID: 34897380) were pooled and integrated using canonical correlation analysis (CCA) in Seurat (Stuart, Cell 2019). All datasets used for analysis were pre-processed using a standard Seurat pipeline: cells containing >200 detected genes, and genes detected in at least 3 cells were included in the analysis using the 'CreateSeuratObject' function with 'min.features = 200' and 'min.cells=3'. Quality control filtering was further performed to remove dead/damaged cells with a high proportion of mitochondrial transcripts, and outlier cells with high UMI numbers (probable doublets/multiplets). However, please note that cutoffs may slightly vary across datasets as QC metrics depend on many technical parameters such as sequencing depth affecting the number of detected UMIs/expressed genes, and variations in mitochondrial transcripts across cell types or differently processed samples. For instance, for mitochondrial transcripts, a <5% cutoff was applied for all datasets, excepted for the data from Williams et al. Nat Immunol 2020 and from Gil-Pulido et al. Cardiovascular Research 2022 where <7.5% and <10% cutoffs were applied, respectively. Pre-processing code for aortic leukocyte datasets can be found as supplemental files of Zernecke et al. Cardiovascular Research 2022 (PMID: 36190844). All data were log normalized using the 'NormalizeData' function in Seurat with default parameters. Data integration was performed using a canonical correlation analysis (CCA) workflow in Seurat with default parameters. After CCA integration, data were scaled using 'ScaleData', principal component analysis performed using 'RunPCA'. Dimensional reduction was performed using 'RunUMAP' with 30 principal components. Clustering was performed using 'FindNeighbors' with 30 principal components, and 'FindClusters' with a resolution of 0.4. Immune cell annotations are based on Zernecke, Circ Res 2020 (PMID: 32673538) and Zernecke, Cardiovascular Research 2022 (PMID: 36190844). We have updated the Methods section to include all these details. As the original Seurat object used to generate previous **Figure 1A** had unfortunately been compromised during backup storage, we had to re-perform the analysis, thus leading to new UMAP embeddings in the revised figures. The new analysis was performed using Seurat v4.3.0.*

*Addressing comment #3 of the reviewer, we identified an issue with cell type annotation in our previous version of **Figure 1A**, likely caused by high background expression of VSMC marker genes (e.g. Acta2, Myh11, Myl9) in a population of double positive CD4⁺CD8⁺ T cells resembling thymic DP T cells (see PMID: 34740961) (new **Supplementary Figure 1 C-D**). This issue was identified in datasets of aortic CD45⁺ cells, presumably due to high levels of ambient RNA from dead cells in these samples and had led to misidentification of this population as VSMCs. We apologize for this error. We now provide a revised **Figure 1A** with updated annotations, and supplemental data (new **Supplementary Figure 1 C-D**) showing expression of lineage marker genes used for identification and annotation of the different*

cell populations. We furthermore provide a clustering analysis with higher granularity, showing subpopulations of macrophages and dendritic cells (see also reply to comment #3 and #4).

We furthermore provide a new supplementary figure showing marker genes used for identification and annotation of cell types (**Supplementary Figure 1 C-D**):

As correctly mentioned by the reviewer, it is possible that some doublets not removed during QC filtering in the initial datasets might also slightly affect the ‘purity’ of clusters, but overall, these technical limitations do not affect our initial conclusion that *Card9* is preferentially detected in macrophages in mouse atherosclerotic aortas. To strengthen this conclusion, we reanalyzed scRNA-seq data from *Ldlr*^{-/-} mouse atherosclerotic aortas from Pan et al, *Circulation* 2020 (PMID: 32962412), where larger numbers of non-immune cells were sampled. Interrogating *Card9* expression in this dataset

corroborates the notion that it is preferentially detected in myeloid cells. These data are now shown in a new **Supplementary Figure 2**. Details regarding this new analysis are provided in the Methods section.

2. Similarly, the methods for analyzing the human coronary artery scRNA-seq data also lacks sufficient details regarding QC, normalization, dimensionality reduction, clustering, and cell type annotation steps.

Response: We agree with the reviewer and now provide additional details in the Methods section. Data from human atherosclerotic coronary arteries (Wirka et al. Nat Med 2019) were analyzed in Seurat v3 (Stuart, Cell 2019) starting from the author provided cell-count matrix (downloaded from Gene Expression Omnibus GSE131778). Cells containing <200 detected genes were excluded, and genes

detected in at least 3 cells were included in the analysis using the 'CreateSeuratObject' functions with 'min.features = 200' and 'min.cells=3'. Further quality control filtering was performed (see illustration figure below, for reviewers only) and cells with >5% mitochondrial transcripts were excluded, as well as cells with outlier number of UMIs (nCount_RNA>15,000). A total of 10,934 cells were analyzed.

As a pre-analysis indicated a substantial patient-driven batch effect, we performed batch correction using Harmony (Korsunsky, Nature Methods 2019) within Seurat, considering each patient as an independent sample. Data were normalized using the 'NormalizeData' function in Seurat with default parameters. 2,000 highly variable genes were identified using 'FindVariableFeatures' (with selection.method = "vst"). Data were scaled using 'ScaleData' with default parameters, and principal component analysis performed using 'RunPCA' with default parameters, and batch corrected using 'RunHarmony' with default parameters. Dimensional reduction was performed using 'RunUMAP(reduction = "harmony", dims = 1:20)', and clustering was performed at a 0.4 resolution using 'FindNeighbors(reduction = "harmony", dims = 1:20)' followed by 'FindClusters(resolution = 0.4)'. Positive marker genes for each cluster were identified using 'FindAllMarkers'. Cell type annotation was performed based on expression of known cell lineage markers, and on cluster annotations in Wirka et al. Nat Med 2019. Please note that the cell type annotation has been slightly revised, and that supporting data are now provided as a supplementary figure (see also reply to comments #6 and #7).

3. In Fig 1a, there is a population of SMCs that seems to be at the intersection of the main SMC and T cell cluster. Since these distinct cell types would be expected to be further from each other in the UMAP space it is possible that this population of SMCs could represent multiplets from the previous publications. Similarly, there is not a single distinct cluster of B cells and the cells closer to myeloid cells might reflect doublets. The authors should carefully inspect the individual datasets before and after integration to reduce this technical artifact.

We thank the reviewer for the attentive examination of the data, which led us to identify an issue in this integrated dataset leading to misidentification of double positive $CD4^+CD8^+$ T cells (see PMID: 34740961) as VSMCs, due to high background expression of VSMC marker genes (e.g. Acta2, Myh11, Myl9) in this cell population in datasets of sorted aortic $CD45^+$ cells, presumably due to high levels of ambient RNA from dead cells in these samples (see also our detailed reply to comment #1). This error does not affect the conclusion that Card9 is preferentially detected in macrophages, which we confirmed using an independent dataset not affected by this technical issue. In our re-analysis of the data presented in **Figure 1A** with a higher clustering resolution, we obtained a clearly distinct cluster of B cells, and a better separation of immune cell subsets.

4. Is CARD9 specifically expressed in anti or pro-inflammatory macrophages? This would be important to discern as the authors make the claim that Card9 is protective and use the broad term of macrophages.

In **Figure 1A**, we now separated aortic macrophages and dendritic cells into the main subsets previously described in scRNA-seq analyses of mouse aortic leukocytes (see Zerneck, *Circ Res* 2020 and Zerneck, *Cardiovasc Res* 2022). In the integrated dataset, we found detectable *Card9* transcripts in resident macrophages, inflammatory macrophages, Trem2/foamy macrophages, type I IFN signaling response macrophages (IFNIC), proliferating macrophages, as well as in monocytes, neutrophils and dendritic cells. We also performed this analysis in individual datasets where total aortic CD45+ were sampled (see Zerneck, *Cardiovascular Research* 2022), corroborating the notion that *Card9* is preferentially detected in myeloid cells including atherosclerosis associated inflammatory and Trem2/Foamy macrophages. Please note that overall, the proportion of macrophages with detectable *Card9* expression is quite low, which is likely an underestimation and might reflect poor capture of *Card9* transcripts in droplet-based scRNA-seq and precludes analyzing *Card9*⁺ and *Card9*⁻ cells separately.

In human data, we also analyzed *CARD9* transcript distribution in a recently published analysis of atherosclerotic vessel mononuclear phagocytes (Zerneck, *Cardiovasc Res* 2022), also showing comparable levels of *CARD9* transcript detection across atherosclerotic plaque macrophage populations. These data are shown in **Supplementary Figure 15 E-G**.

5. Given that the authors are highlighting the macrophage specific role of CARD9, it would be helpful to show specific macrophage markers that are expressed within the large cluster to support the cell type annotations.

*Response: We thank the reviewer for the suggestion and now included a supplementary figure showing expression of marker genes used to identify cell lineages, including macrophage markers (see reply to comment #1 and new **Supplementary Figure 1**).*

6. In Fig 8, it is confusing that there are two separate clusters labeled as VSMCs that are away from each other. The authors should further inspect these to determine whether these represent SMCs and pericytes separately.

*In the previous version of this figure, we had, for simplification purposes, grouped some clusters together under a single annotation, and had indeed grouped mural cells (pericytes and VSMCs) together under the “VSMC” annotation. Likewise, 2 subclusters of fibroblasts can be distinguished and had been grouped together under the ‘Fibro/MSC’ annotation. In the revised version of **Figure 7F**, all these subclusters are shown.*

Figure 7F: updated UMAP plot showing mural cells (VSMC and pericytes) and fibroblast/MSC subclusters.

*We now provide supporting data showing the expression of specific marker genes used for cell type identification and annotation (see below and new **Supplementary Figure 15**).*

7. Similar to Fig 1, it would be helpful to show markers of human macrophages that are expressed within the large cluster to support the cell type annotations. This could be done with UMAP feature plots or included within Supplementary tables.

We now provide data supporting cell type annotations, including specific monocyte and macrophage markers (CD14, C1QC, C5AR1 encoding CD88, TREM2) in Supplementary Figure 15 (see also reply to comment #6)

8. Regarding the immunostaining in Fig 8, this appears incomplete and there are no general histology stains to assess the lesion stage or location of the staining in D and E. Some quantitation of CARD9 mRNA or protein levels in human athero versus normal arteries would help support the staining results.

Response: To address the reviewer's concern, in collaboration with Prof. G Pasterkamp's group (Utrecht University), we analyzed by Western Blot, CARD9 expression in atheromatous versus fibrous human plaques. We found higher CARD9 protein expression in atheromatous lesions. Given that the number of samples was limited, these data have been inserted as supplementary figure 14B.

Minor comments

1. Some of the text labels in Fig3, Fig 6 and Fig7 are quite small and almost not legible. The text should be increased to be consistent with the other figures.

Response: The modifications have been done accordingly

2. Magnification or ruler should be added to the immunostaining images in Fig 8.

Response: The modifications have been done accordingly

REVIEWERS' COMMENTS

Reviewer #1 (Remarks to the Author):

The authors have addressed all my comments and the manuscript has increased in quality.

Reviewer #2 (Remarks to the Author):

The authors thoroughly addressed all of my previous comments and the manuscript has substantially improved. Therefore, I recommend to accept this innovative and important translational atherosclerosis study to be published by Nature Communications.

Reviewer #3 (Remarks to the Author):

The authors have sufficiently addressed my comments and the manuscript is much improved. I have no further comments.

Responses to reviewers' comments

Reviewer #1 (Remarks to the Author):

The authors have addressed all my comments and the manuscript has increased in quality.

Response: We would like to thank the reviewer for his/her comments that helped to improve the manuscript

Reviewer #2 (Remarks to the Author):

The authors thoroughly addressed all of my previous comments and the manuscript has substantially improved. Therefore, I recommend to accept this innovative and important translational atherosclerosis study to be published by Nature Communications.

Response: We would like to thank the reviewer for his/her comments that helped to improve the manuscript

Reviewer #3 (Remarks to the Author):

The authors have sufficiently addressed my comments and the manuscript is much improved. I have no further comments.

Response: We would like to thank the reviewer for his/her comments that helped to improve the manuscript